

**Biodiversity and trophic ecology of hydrothermal vent fauna associated with**
**tubeworm assemblages on the Juan de Fuca Ridge**
Yann Lelièvre[1,2*], Jozée Sarrazin[1], Julien Marticorena[1], Gauthier Schaal[3], Thomas Day[1], Pierre
Legendre[2], Stéphane Hourdez[4], Marjolaine Matabos[1]
[1]*Ifremer, Centre de Bretagne, REM/EEP, Laboratoire Environnement Profond, 29280*
*Plouzané, France*
[2]*Département de sciences biologiques, Université de Montréal, C.P. 6128, succursale Centre-*
*ville, Montréal, Québec, H3C 3J7, Canada*
[3]*Laboratoire des Sciences de l'Environnement Marin (LEMAR), UMR 6539 9*
*CNRS/UBO/IRD/Ifremer, BP 70, 29280, Plouzané, France*
[4]*Sorbonne Universités, UPMC Univ. Paris 06, CNRS UMR 7144, Adaptation et Diversité en*
*Milieu Marin, Station Biologique de Roscoff, 29688 Roscoff, France*
**Corresponding author**
Lelièvre Yann
(1) Département de sciences biologiques, Université de Montréal, C.P. 6128, succursale
Centre-ville, Montréal, Québec, H3C 3J7, Canada; (2) Ifremer, Centre de Bretagne, REM/EEP,
Laboratoire Environnement Profond, 29280 Plouzané, France.
Telephone number in Canada: +1 514 343 7591
Email address: yann.lelievre@ifremer.fr



**Abstract**
Hydrothermal vent sites along the Juan de Fuca Ridge in the north-east Pacific host dense
populations of *Ridgeia piscesae* tubeworms that promote habitat heterogeneity and local
diversity. A detailed description of the biodiversity and community structure is needed to
help understand the ecological processes that underlie the distribution and dynamics of
deep-sea vent communities. Here, we assessed the composition, abundance, diversity and
trophic structure of six tubeworm assemblages, corresponding to different successional
stages, collected on the Grotto hydrothermal edifice (Main Endeavour, Juan de Fuca Ridge)
at 2196 m depth. Including *R. piscesae*, a total of 36 macrofaunal taxa were identified to the
species level. Although polychaetes made up the most diverse taxon, faunal densities were
dominated by gastropods. Most tubeworm aggregations were numerically dominated by the
polychaete *Amphisamytha carldarei* and gastropods *Lepetodrilus fucensis* and *Depressigyra*
*globulus*. The highest diversities were found in mature tubeworm aggregations,
characterized by fairly long tubes. The high biomass of grazers and the high resource
partitioning at small scale illustrates the importance of the diversity of free-living microbial
communities in the maintenance of the food web. Although symbiont-bearing invertebrates
*R. piscesae* represented a large part of the total biomass, the absence of specialized
predators on this potential food source suggests that its primary role lies in community
structuring. Vent food webs did not appear to be organized through predator-prey
relationships. For example, although trophic structure complexity increased with ecological
successional stages, showing a higher number of predators in the last stages, the food web
structure itself did not change across assemblages. We suggest that environmental gradients
provided by the biogenic structure of tubeworm bushes generate a multitude of ecological
niches and contribute to the partitioning of nutritional resources, releasing communities
from competition pressure for resources, thus allowing species co-existence.

*Keywords: Juan de Fuca Ridge; hydrothermal vents; Ridgeia piscesae; community structure;*
*diversity; stable isotopes; food webs.*





## 1. Introduction

Deep-sea hydrothermal vents have developed along mid-ocean ridges and back-arc spreading centres, which are characterized by strong volcanic and tectonic activity. The resulting hydrothermal fluid fosters dense communities of highly specialized fauna that colonize the steep physical and chemical gradients created by the mixing of hot vent fluids with cold seawater. These communities are distributed according to species' physiological tolerance (Childress and Fisher, 1992; Luther et al., 2001), resource availability (De Busserolles et al., 2009; Levesque et al., 2003) and biotic interactions (Lenihan et al., 2008; Micheli et al., 2002; Mullineaux et al., 2000, 2003). Although the fauna are highly dissimilar between oceanic basins (Bachraty et al., 2009; Moalic et al., 2011), hydrothermal communities throughout the world share some ecological similarities including a food web based on chemosynthesis (Childress and Fisher, 1992), low species diversity compared with adjacent deep-sea and coastal benthic communities (Van Dover and Trask, 2000; Tunnicliffe, 1991), high levels of endemism (Ramirez-Llodra et al., 2007), and elevated biomass associated with the presence of large invertebrate species.

The high spatial heterogeneity of environmental conditions in vent ecosystems is amplified by stochastic or periodic temporal variation in hydrothermal activity, influencing the composition (Sarrazin et al., 1999), structure (Marcus et al., 2009; Sarrazin et al., 1997; Tsurumi and Tunnicliffe, 2001) and dynamics (Lelièvre et al., 2017; Nedoncelle et al., 2013, 2015; Sarrazin et al., 2014) of faunal communities. In addition, the complexity of vent habitats is increased by engineer species, whose presence strongly contributes to the modification of the physical (temperature, hydrodynamics processes) and chemical (hydrogen sulfide, methane, oxygen, metals and other reduced chemicals) properties of the environment either by creating three-dimensional biogenic structures (autogenic species) or through their biological activity (allogeneic species) (Jones et al., 1994, 1997). Habitat provisioning and modification by engineer species increases the number of potential ecological niches and, consequently, influences species distribution and contributes to the increase in local diversity (Dreyer et al., 2005; Govenar and Fisher, 2007; Urcuyo et al., 2003). Engineer species promote local diversity through various ecological mechanisms (Bergquist et al., 2003), providing secondary substratum for colonization, a refuge from



predation and unfavourable abiotic conditions or important food sources that enhance the
development of macro- and meiofaunal communities (Dreyer et al., 2005; Galkin and
Goroslavskaya, 2010; Gollner et al., 2006; Govenar et al., 2005, 2002; Govenar and Fisher,
2007; Turnipseed et al., 2003; Zekely et al., 2006).

Hydrothermal vent food webs are mainly based on local microbial chemosynthesis (Childress
and Fisher, 1992), performed by free-living or/and symbiotic chemoautotrophic
microorganisms that utilize the chemical energy released in the oxidation of reduced
chemicals species (e.g. $H_2S$, $CH_4$) present in the hydrothermal fluids (Childress and Fisher,
1992). Several electron donors (e.g. $H_2$, $H_2S$, $CH_4$, $NH_4^+$, etc.) and electron acceptors (e.g. $O_2$,
$NO_3^-$, $SO_4^{2-}$, etc.) can be used by these microorganisms as energy sources, converting
inorganic carbon (e.g. $CO_2$) into simple carbohydrates (Fisher et al., 2007). Chemosynthetic
primary production is exported to the upper trophic levels through direct ingestion (primary
consumers), or through the presence of intra- or extracellular symbiosis. Upper trophic
levels (secondary consumers) are represented by local predators and scavengers feeding on
primary consumers and by abyssal species attracted by the profusion of food. Although
behavioural observations and stomach content analyses remain limited in these remote
deep-sea habitats, stable isotope analyses are widely used to study faunal trophic
interactions in these environments (Conway et al., 1994). The emergence of isotopic
methods has opened new perspectives in the understanding of food-web functioning and
the organization of species diversity within hydrothermal ecosystems around the globe
(Bergquist et al., 2007; De Busserolles et al., 2009; Van Dover, 2002; Erickson et al., 2009;
Gaudron et al., 2012; Levesque et al., 2006; Levin et al., 2009; Limén et al., 2007; Portail et
al., 2016; Soto, 2009; Sweetman et al., 2013). The carbon isotope composition ($\delta^{13}C$) is an
indicator of the food assimilated and remains relatively constant during trophic transfers
(± 1‰). The kinetics of enzymes involved in the biosynthetic pathways of autotrophic
organisms influence the carbon isotope ratio ($^{13}C/^{12}C$), allowing the discrimination between
the sources fuelling the community (Conway et al., 1994; Van Dover and Fry, 1989). Nitrogen
isotope composition ($\delta^{15}N$) provides information on trophic levels (Michener and Lajtha,
2008) and becomes enriched in heavy isotopes at a rate of ± 3.4‰ at each trophic level. At
the community scale, $\delta^{13}C$ and $\delta^{15}N$ signatures of all species in the ecosystem are used to
retrace carbon and nitrogen fluxes along the trophic network and, therefore, to reconstitute

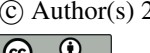



the food web (Levin and Michener, 2002). Despite the relatively low diversity of the deep-
sea community, ample evidence suggests that the deep-sea hydrothermal food-web
structure is complex (Bergquist et al., 2007; Portail et al., 2016) including many trophic guilds
(Bergquist et al., 2007; De Busserolles et al., 2009) and multiple sources of primary
production (Van Dover and Fry, 1994).

Active hydrothermal vents on the Juan de Fuca Ridge (north-east Pacific) are colonized by
populations of the siboglinid polychaete *Ridgeia piscesae* (Urcuyo et al., 2003) forming dense
faunal assemblages in areas of high to low fluid flux activity. Diverse heterotrophic faunal
species inhabit these tubeworm bushes, with a dominance of polychaete and gastropod
species (Bergquist et al., 2007; Govenar et al., 2002; Marcus et al., 2009; Tsurumi and
Tunnicliffe, 2001, 2003). To date, few studies have described the communities associated
with the *R. piscesae* tubeworm assemblage of the Main Endeavour vent field, either in terms
of diversity (Bergquist et al., 2007; Sarrazin et al., 1997) or trophic ecology (Bergquist et al.,
2007). Six distinct faunal assemblages exhibiting patchy distributions have been identified on
the Smoke & Mirrors hydrothermal edifice, and represent different successional stages
(Sarrazin et al., 1997). Assemblages I and II are characterized by pioneer *Paralvinella*
*palmiformis* polychaete species that colonize new unstable high-temperature surfaces and
whose biological activity tends to stabilize the substratum (Juniper et al., 1992; Sarrazin et
al., 1997). Assemblage III is marked by dense aggregations of *P. palmiformis* and the
colonization of high densities of gastropods *Lepetodrilus fucensis*, *Depressigyra globulus* and
*Provanna variabilis* (Sarrazin et al., 1997). Assemblage IV is characterized by the growth of *R.*
*piscesae*, leading to assemblage V associated with a more complex physical structure and
consequently with an increase in local diversity, density and biomass (Bergquist et al., 2003;
Sarrazin et al., 1997; Tsurumi and Tunnicliffe, 2003). Finally, assemblage VI characterizes a
senescent phase in which *R. piscesae* gradually dies and its associated species disappear,
with a dominance of filamentous bacteria and detritivores (Sarrazin et al., 1997). A
successional model proposes that the transition between the first two assemblages is mostly
driven by biotic interactions, and those between the other assemblages are principally
initiated by modifications in hydrothermal activity (Sarrazin et al., 1997). Within a single
assemblage of *R. piscesae* tubeworm from diffuse flow vent environments of Easter Island



(Main Endeavour, Juan de Fuca Ridge), Bergquist *et al*. (2007) reported that tubeworm-
generated habitats supported a diverse community, with a complex local food web.

Since 2011, a camera installed on the *Ocean Networks Canada* cabled observatory has been
recording high-resolution imagery of a *R. piscesae* tubeworm assemblage and its associated
fauna on the active Grotto hydrothermal edifice (Main Endeavour, Juan de Fuca Ridge). The
processing of this data provided new insights on the influence of astronomic and
atmospheric forcing on vent faunal dynamics (Lelièvre et al., 2017), but thorough knowledge
of the faunal communities observed by the camera is still needed to understand and
interpret the temporal patterns and their underlying mechanisms. However, although video
imagery is useful for investigating the spatial distribution of communities (Cuvelier et al.,
2011; Sarrazin et al., 1997), species behaviour (Grelon et al., 2006; Matabos et al., 2015) and
temporal dynamics of a sub-set of the species (Cuvelier et al., 2014; Lelièvre et al., 2017),
direct sampling is an essential and complementary approach for determining faunal
composition, abundance and species diversity and functioning (Cuvelier et al., 2012). In this
context, the objectives of the present study were: (*i*) to identify the composition and
structure of six faunal assemblages associated with *R. piscesae* tubeworm bushes on the
Grotto hydrothermal edifice, specifically with respect to density, biomass and species
diversity; (*ii*) to characterize the trophic structure of these biological communities and (*iii*) to
assess how diversity and trophic relationship vary over the different successional stages.

## 167 2. Materials and Methods

### 168 2.1. Geological setting

The Juan de Fuca Ridge (JdFR) (Fig. 1a) is an intermediate spreading-rate ridge between the
Pacific and Juan de Fuca plates in the north-east Pacific Ocean. The Endeavour Segment
(47°57'N, 129°06'W) (Fig. 1b) constitutes a ~ 90 km long section of the JdFR, bounded to the
north by the Middle Valley site and to the south by the Cobb Segment. It is characterized by
a 500–1000 m wide axial valley whose walls reach up to 200 m in height (Delaney et al.,
1992). The five major vent fields – Sasquatch, Salty Dawg, High Rise, Main Endeavour (MEF)
and Mothra – found on the Endeavour axial valley are separated by 2–3 km.



The MEF (Fig. 1c) is the most intense and active of the five hydrothermal fields, with the
presence of high-temperature (370-390°C), actively venting sulfide edifices and diffuse low-
temperature (10-25°C) venting areas (Delaney et al., 1992; Kelley et al., 2012). Within the
MEF, Grotto (47°56.958'N, 129°5.899'W, Fig. 1d) is an active hydrothermal sulfide vent
cluster (15 m long by 10 m wide by 10 m high) located at 2196 m depth that forms an open
cove to the north. This edifice is characterized by high short-term variation in heat flux, but a
relative stability in years with low seismic activity (Xu et al., 2014). Like many other MEF
hydrothermal edifices, the site is largely colonized by dense assemblages of *R. piscesae*
(Polychaeta, Siboglinidae) with their associated fauna (Sarrazin et al., 1997).

**2.2. Faunal assemblage sampling**
Sampling took place during the ONC oceanographic cruises *Wiring the Abyss 2015 and 2016*
from 25 August to 14 September 2015 on the R/V *Thomas G. Thompson*, and from 10 May to
29 May 2016 on the E/V *Nautilus*, respectively. Using the remotely operated vehicles (ROVs)
*Jason* and *Hercules*, three assemblages of *R. piscesae* tubeworms and their associated fauna
were sampled each year at different locations on the Grotto hydrothermal edifice (n=6; S1 to
S6, Fig. 2). For each sample, a checkerboard of 7 x 7 mm squares was first placed on each
tubeworm assemblage to estimate the surface area. Then, the first suction sample was
taken to recover the mobile fauna, followed by collection of tubeworms and their associated
fauna, which were placed in a "bio-box" using the ROV's mechanical arm. A final suction
sample on the bare surface was performed to recover the remaining fauna. The final
sampled surface was filmed with the ROV camera to estimate its surface using imagery (see
protocol in Sarrazin *et al.* 1997 (Sarrazin et al., 1997)).

**2.3. Sample processing**
*2.3.1. Sample processing and identification*
On board, all faunal samples were washed over stacked sieves (250 µm, 63 µm and 20 µm
mesh sizes). Macrofaunal specimens (>250 µm) were preserved in 96 % ethanol and
meiofauna (<63 µm) in 10 % seawater formalin. In the laboratory, bushes of *R. piscesae* were
thoroughly disassembled and each tube was washed and sieved a second time. All
associated macrofaunal organisms were sorted, counted and identified to the lowest





possible taxonomic level. Specimens whose identification was unclear were sent to experts
for identification and/or description. When available, trophic guilds from the literature
(symbiont host, bacterivore, scavenger/detritivore or predator) were assigned to each vent
species. For species with unknown diets, the assignment was based on trophic guilds
identified from closely related species (within the same family).

### 2.3.2 Habitat complexity and biomass

For each tubeworm assemblage, the density measured in number of individuals per square
meter (ind m$^{-2}$) was calculated. In addition to the surface they occupy, *R. piscesae* tubes
create a three-dimensional (3D) structure for other vent animals to colonize. An estimation
of the volume of each assemblage provided a proxy for habitat complexity. For this, in each
sample, 10 % of the tubeworm tubes were randomly selected and measured. Assuming that
the tubes are erected vertically, sampling volume was estimated by multiplying the mean
tube length by the sampled surface area. Final densities are therefore expressed per m$^{3}$ to
account for this 3D space. Biomass estimates were obtained for a random sample of 3 to 10
individuals of each species. The total dry mass (DM) of each species corresponds to the mass
obtained after drying each individual at 80°C for 48 h; the ash-free dry mass (AFDM) was
obtained after combustion in a muffle furnace at 500°C for 6 h. Absolute biomass of each
species was calculated by multiplying the relative biomass by the abundance of each species.

### 2.3.3. Stable isotope processing

Sample preparation for stable isotope analyses was specimen size-dependent. For large
specimens, muscle tissue was dissected and used for stable isotope analyses. In the case of
intermediate-size specimens, the gut content was removed. For small taxa, entire individuals
were analysed or pooled to reach the minimum required mass for isotopic analysis. Samples
were freeze-dried and ground into a homogeneous powder using a ball mill or agate mortar.
About 1.3-1.4 mg of the powder was precisely measured in tin capsules for isotope analysis.
For species containing carbonates (i.e. gastropods, ostracods, amphipods, etc.), individuals
were acidified to remove inorganic carbon. Acidification was carried out by the addition of
0.1 M HCl. The sample was then dried at 60°C for 24 h under a fume extractor to evaporate
the acid. Five replicates per species were analysed. Carbon and nitrogen isotope ratios were



determined using a Thermo Scientific FLASH EA 2000 elemental analyser coupled with a
Thermo Scientific Delta V Plus isotope ratio mass spectrometer. Values are expressed in δ
(‰) notation relative to Vienna Pee Dee Belemnite and atmospheric $N_2$ as international
standards for carbon and nitrogen, respectively, according to the formula: $\delta^{13}C$ or $\delta^{15}N$ =
[($R_{sample}/R_{standard}$)-1] x $10^3$ (in ‰) where R is $^{13}C/^{12}C$ or $^{15}N/^{14}N$. Analytical precision based on
repeated measurements of the same sample was below 0.3‰ for both $\delta^{13}C$ and $\delta^{15}N$.

**2.4. Statistical analyses**
In the present study, *R. piscesae* was regarded as a habitat builder and thus discarded from
the statistical analyses. Species-effort curves were computed for each faunal sample
collected to assess the robustness of the sampling effort. Local diversity (i.e. α diversity) was
estimated for each tubeworm assemblage from several complementary indices (Gray, 2000)
using the vegan package in R (Oksanen et al., 2017): species richness (S), exponential
Shannon entropy (D), Simpson's (1-λ') indices of species diversity and Pielou's evenness
index (J').

**3. Results**
**3.1. Species-effort curves, tubeworm complexity and diversity**
The rarefaction curves (Fig. 3) showed that, overall, the collected samples (S1 to S6) gave a
fairly good representation of the species diversity on the Grotto hydrothermal edifice. In
2015, sample S2 (24 taxa, excluding *R. piscesae*) and S3 (31 taxa) rarefaction curves seemed
to reach a plateau. S1 cumulated a total of 28 macrofaunal taxa. The samples from year
2016 exhibited lower species richness and did not reach an asymptote. Samples S4 and S5
had a macrofaunal species richness of 19 taxa, while only 14 taxa were found in sample S6
(Fig. 3).

The volumes of the samples were used as an approximate measure of habitat complexity of
the 3D structures of the *R. piscesae* assemblages. Samples S1 and S3 showed similar
patterns, with sampling surfaces of 12.36 and 11.92 $dm^2$ and mean tube lengths of
17.24 ± 6.38 and 17.89 ± 5.69 cm, respectively (Table 1). Therefore, S1 and S3 were
characterized by a similar degree of complexity, with a volume of 21.31 and 21.33 $dm^3$.





Sample S2 displayed a sampling area of less than half of that of S1 and S3 (6.33 dm$^2$) and a
mean tube length of 8.16 ± 2.14 cm with an estimated resulting volume of 5.16 dm$^3$.
Samples S4 to S6 were substantially smaller than S1, S2 and S3, with a sampling surface
between 1.22 and 1.59 dm$^2$. *R. piscesae* tubes were short in samples S4 and S5 leading to a
sampling volume of 0.7 and 0.86 dm$^3$ respectively (Table 1). Sample S6 displayed tube
lengths similar to S2 leading to a sampling volume of 1.02 dm$^3$ (Table 1).

Alpha diversity measures showed that S3 displayed the highest diversity (Shannon D = 6.053;
1-λ' = 0.778), slightly greater than S2 (D = 5.398; 1-λ'= 0.749) and S1 (D = 5.377; 1-λ'= 0.728)
(Table 1). The lowest diversity values were observed in S5 (D = 4.348; 1-λ' = 0.697), S6
(D = 3.998; 1-λ' = 0.633) and S4 (D = 2.605; 1-λ' = 0.550). The S2 and S3 samples showed a
more even distribution (J') of individuals among taxa than the other assemblages. In
contrast, S4 had the lowest evenness (J'= 0.325) (Table 1). Species richness was significantly
correlated with *R. piscesae* tube length ($R^2_{adj}$ = 0.60, *p-value* = 0.042).

**3.2. Composition and structure of Grotto vent communities**
The species lists and abundances for each sample collected within the Grotto hydrothermal
edifice are provided in Table 2. A total of 148 005 individuals representing 35 macrofaunal
taxonomic groups were identified in the six *R. piscesae* assemblages (S1 to S6) sampled on
the Grotto edifice. Overall, gastropods (5 taxa) and polychaetes (19 taxa) respectively
accounted for 61.51 ± 16.9 % and 29.06 ± 13.06 % of the total macrofaunal abundance. The
numerically most abundant species were the gastropods *L. fucensis* and *D. globulus* as well
as the polychaete *Amphisamytha carldarei* representing respectively 33.95 ± 7.58 %,
24.54 ± 15.68 % and 15.08 ± 13.57 % of the total abundance. The highest macrofaunal
densities were observed in samples S4 (19 364 286 ind m$^{-3}$), S5 (7 461 628 ind m$^{-3}$), S2
(5 196 318 ind m$^{-3}$) and S3 (3 143 241 ind m$^{-3}$), whereas S6 and S1 had the lowest densities
with 1 607 843 ind m$^{-3}$ and 1 523 932 ind m$^{-3}$, respectively. The foundation species *R.*
*piscesae* represented a large part of the total biomass, with a mean of 69.3 ± 15.7 % of the
total biomass, followed by the gastropods *L. fucensis* (14.96 ± 4.05 %) and *D. globulus* (6.76 ±
8.34 %). A high percentage (30 %) of the species were only found in 1 or 2 samples.




More specifically, S1 was dominated by gastropod species such as *L. fucensis* (617 785 ind m⁻
³; 12.48 % of total biomass), *D. globulus* (156 265 ind m⁻³; 1.12 % of total biomass) and *P.*
*variabilis* (27 452 ind m⁻³; 2.25 % of total biomass) (Table 2). High densities contrasted with
low biomass were also observed for the ampharetid polychaete *A. carldarei* and the syllid
polychaete *Sphaerosyllis ridgensis*. S2 was also dominated by *L. fucensis*, *D. globulus* and *A.*
*carldarei*, with, however, a high proportion of ostracods *Euphilomedes climax* (475 388
ind m⁻³; 0.10 % of total biomass) (Table 2). S3 was largely dominated by *A. carldarei*
(1 073 511 ind m⁻³; 1.84 % of total biomass) and, to a lesser extent, was almost equally
dominated by *L. fucensis* (702 438 ind m⁻³; 11.53 % of total biomass) and *D. globulus*
(619 784 ind m⁻³; 2.86 % of total biomass). Polychaetes were also dominant, with the
presence of *S. ridgensis* (52 414 ind m⁻³; <0.001 % of total biomass), the dorvilleid
*Ophryotrocha globopalpata* (40 319 ind m⁻³; <0.001 % of total biomass) and the maldanid
*Nicomache venticola* (7079 ind m⁻³; 1.09 % of total biomass). There were high densities of *P.*
*variabilis* (138 678 ind m⁻³; 6.78 % of total biomass), the solenogaster *Helicoradomenia juani*
(172 011 ind m⁻³; 0.14 % of total biomass), the acarida *Copidognathus papillatus*
(150 633 ind m⁻³; <0.001 % of total biomass), the ostracod *Xylocythere sp. nov*.
(82 419 ind m⁻³; <0.001 % of total biomass) and the pycnogonid *Sericosura verenae* (22 644
ind m⁻³; 0.84 % of total biomass) (Table 2). S4 was dominated by *L. fucensis* (7 700 000 ind m⁻
³; 19.43 % of total biomass) and *D. globulus* (9 195 714 ind m⁻³; 13.02 % of total biomass)
and, to a lesser extent, by the alvinellid polychaete *P. palmiformis* (591 429 ind m⁻³; 6.78 %
of total biomass) (Table 2). S5 was also dominated by *L. fucensis* and *D. globulus*, followed by
*E. climax* (624 419 ind m⁻³; <0.001 % of total biomass) and *P. variabilis* (594 186 ind m⁻³; 10 %
of total biomass) (Table 2). Finally, S6 was also dominated *L. fucensis* and *D. globulus* and, to
a lesser extent, by *A. carldarei* (86 275 ind m⁻³; 0.07 % of total biomass) and the alvinellid
polychaete *Paralvinella pandorae* (63 725 ind m⁻³; <0.001 % of total biomass) (Table 2).

**3.4. δ¹³C and δ¹⁵N isotopic composition**
δ¹³C values of the vent fauna ranged from −33.4 to −11.8 ‰ among the different samples
(Fig. 4). More specifically, δ¹³C values ranged from −33.4 to −13.5 ‰ for S1, from −33.4 to
−15.4 ‰ for S2 and from −32.4 to −14.7 ‰ for S3. Samples from S4, S5 and S6 displayed
slightly narrower δ¹³C ranges, varying from −30.3 to −12.5 ‰, from −31.3 to −14.8 ‰ and



from −32.3 to −11.8 ‰, respectively; most species were enriched in $^{13}$C relative to the S1, S2
and S3 samples (Fig. 4). Overall, the gastropod *P. variabilis* (species #2) was the most
depleted in $^{13}$C with values around −32.2‰ (± 1.2 ‰). In contrast, *R. piscesae* siboglinids
(species #1) showed the highest $\delta^{13}$C values, with constant values around −14.7 ‰ (±
1.0 ‰). The range of $\delta^{15}$N values in faunal assemblages varied between −8.5 and 9.4 ‰ (Fig.
4). More specifically, S1 values ranged from 0.3 to 8.4 ‰, S2 from 0.4 to 9.2‰, S3 from −2.7
to 8.3‰, S4 from −1.3 to 8.7 ‰, S5 from −1.1 to 6.4 ‰ and S6 from −8.5 to 9.4 ‰ (Fig. 4).
Overall, 15 species showed a $\delta^{15}$N > 5 ‰ in S1, S2 and S3 assemblages but only 4 species
were over 5 ‰ in $\delta^{15}$N in S4, S5 and S6. In contrast to their $\delta^{13}$C values, *P. variabilis* and *R.*
*piscesae* displayed similar and relatively stable $\delta^{15}$N values among samples with 0.3 ‰
(± 0.8 ‰) and 1.5 ‰ (± 1.1 ‰), respectively.

### 3.5. Biomass distribution in the Grotto trophic network

The projection of the species isotopic ratios weighted by biomass is useful for estimating the
relative contributions of the different trophic pathways within the vent assemblages (Fig. 5).
In our study, there were similar patterns of biomass distribution in the six sampled
assemblages. In all samples, the engineer polychaete *R. piscesae* (species #1) represented
the highest biomass (69.3 ± 16 %). It was considered to be a structuring species of our vent
ecosystem and was not included in the following biomass distribution analysis. With a
biomass ranging from 78.9 to 95.8 % (89.6 ± 6.8 %), gastropods seemed to play an important
role in the trophic food web of communities associated with the siboglinid tubeworms. The
gastropod biomass was dominated by *L. fucensis* (species #4), which accounted for 31.5 to
82.8 % (55.8 ± 18.3 %) of the total biomass. In addition to *L. fucensis*, the gastropods *D.*
*globulus* (species #3), *P. variabilis* (species #2) and *Buccinum thermophilum* (species #5)
showed relatively high biomass within the different samples, ranging from 5.6 to 36.6 %
(16.5 ± 13.8 %), 0.6 to 26.3 % (10.9 ± 9.8 %) and 0 to 16.1 % (6.4 ± 6.8 %), respectively (Fig.
5). However, in some assemblages, other species also significantly contributed to the total
biomass. For example, in S3, the polychaete *A. carldarei* (species #7) contributed
substantially (7.2 %) to the total biomass. Similarly, in S4, the polychaete *P. palmiformis*
(species #13) contributed to 16.4 % of the total biomass. Our results also show that the
biomass declined from the bacterivore to the predator guilds in the Grotto trophic network.





## 4. Discussion

### 4.1. Communities and diversity

Hydrothermal ecosystems of the north-east Pacific are dominated by dense populations of tubeworms *R. piscesae*. In this study, a total of 36 macrofaunal taxonomic groups (including *R. piscesae*) were found in the six tubeworm assemblages sampled on the Grotto edifice, which is consistent with previous community knowledge in the region (Bergquist et al., 2007). In this study, macrofaunal species richness was slightly lower than that observed at the Easter Island hydrothermal site on the Main Endeavour Field, where a total of 39 species had been identified in a single *R. piscesae* bush (Bergquist et al., 2007). Another study also reported the presence of 39 macrofaunal species in 25 collections from the Axial Segment (JdFR), but lower values have been reported on other segments, with 24 species in 7 collections from the Cleft Segment (JdFR) and 19 species in 2 collections from the CoAxial Segment (JdFR) (Tsurumi and Tunnicliffe, 2003). These levels of diversity are lower than that found in *Riftia pachyptila* bushes on the East Pacific Rise, where species richness in 8 collections reached 46 taxa (Govenar et al., 2005). Macrofaunal diversity was also lower than those obtained in engineer mussel beds from Lucky Strike on the Mid-Atlantic Ridge, with 41 taxa identified (Sarrazin et al., 2015), or from the northern and southern East Pacific Rise, with richnesses of 61 and 57 taxa, respectively (Van Dover, 2003). Faunal dissimilarities between worldwide hydrothermal ecosystems may be closely related to the geological context (ridge, back-arc basins), history of species colonization, connectivity to neighbouring basins, presence of geographic barriers (transform faults, hydrodynamic processes, depths, etc.), stability of hydrothermal activity, age of the vent system and inter-site distances (Van Dover et al., 2002). Discrepancies in sampling effort may also account for variation between sites and regions.

*R. piscesae* tubeworm assemblages sampled on the Grotto edifice were characterized by the dominance of a few species (e.g. *L. fucensis*, *D. globulus*, *A. carldarei*), a pattern that has also been reported from other hydrothermal sites of the world oceans: Mid-Atlantic Ridge (Cuvelier et al., 2011; Sarrazin et al., 2015), East Pacific Rise (Govenar et al., 2005), JdFR (Sarrazin and Juniper, 1999; Tsurumi and Tunnicliffe, 2001) and the southern East Pacific Rise (Matabos et al., 2008). Polychaetes were the most diverse taxa, representing half of the





macrofaunal species richness (19 taxa). Similar results have been reported within *R. piscesae*
bushes on Easter Island, with the identification of 23 polychaete species (Bergquist et al.,
2007). Although the dominant species were similar among samples, variation between
samples involved mainly the relative abundance of the few dominant species and the
identity of the rare species. These variations may result from differences in sampling
strategy. The areas sampled in 2016 were smaller than in 2015 and a problem with the
sampling boxes may have led to the loss of some individuals. Variation in species richness
and diversity among samples may also depend on the presence of environmental gradients,
created by the mixing between ambient seawater and hydrothermal effluents (Sarrazin et
al., 1999). Unfortunately, no environmental data were recorded with our samples. However,
physical and chemical conditions are known to change along the ecological succession
gradient on the MEF from newly opened habitat characterized by high temperature and
sulfide concentrations, colonized by the sulfide worm *Paralvinella sulfincola*, to mature
communities in low diffuse venting areas characterised by low temperatures and sulfide
concentrations and colonized by long skinny *R. piscesae* tubeworms (Sarrazin et al., 1997).
Tubeworm assemblages S1 and S3 were visually recognized as type V low-flow assemblages
(Sarrazin et al., 1997), characterized by a mature phase of *R. piscesae* bush development and
a high level of complexity. This assessment was confirmed by the length of the collected
tubes (17 cm on average). Both assemblages showed the highest species richness, diversity
and most complex trophic network, illustrating the strong influence of engineer species and
the importance of biogenic structure in the diversification and persistence of the local
resident fauna. By increasing the number of micro-niches available for vent species, the 3D
structure of *R. piscesae* bushes helps to increase the environmental heterogeneity and
thereby promotes species richness and diversity at community scales (Jones et al., 1997;
Tsurumi and Tunnicliffe, 2003). As mentioned by several authors (Bergquist et al., 2003;
Govenar et al., 2002; Tsurumi and Tunnicliffe, 2003), various ecological mechanisms may
explain the influence of *R. piscesae* tubeworms on local diversity: new habitats generated by
tubeworm bushes provide (*i*) a substratum for attachment and colonization; (*ii*) interstitial
spaces among intertwined tubes, increasing habitat gradients and therefore the number of
ecological niches; (*iii*) a refuge to avoid predators and to reduce the physiological stress
related to abiotic conditions and (*iv*) a control on the transport of hydrothermal vent flow
and nutritional resource availability. Assemblages S2 and S5, also identified as type V low-



flow assemblages (Sarrazin et al., 1997), presented shorter tube lengths than S1 and S3,
which might explain the lower species richness in these two samples. Polychaete densities
on Grotto were dominated by the ampharetid *A. carldarei* (89.9 ± 2.8 %, not including S4 and
S6). High densities in *R. piscesae* assemblages may be related to the specificity of this family
with high ecological tolerance to environmental conditions and, therefore, to their ability to
take advantage of a wide range of ecological niches (McHugh and Tunnicliffe, 1994). Similar
to *L. fucensis*, *A. carldarei* is characterized by early maturity and high fecundity, contributing
to the success of this species in vent habitats (McHugh and Tunnicliffe, 1994). The
dominance of gastropods *L. fucensis* and *D. globulus* as well as the relatively high presence
of the *Paralvinella* polychaete species in samples S4 and S6 suggest that they belong to
lower succession levels, corresponding to transitory states between types III and IV
assemblages (Sarrazin et al., 1997). The latter two samples were characterized by low
species richness and diversities. We hypothesize that the numerical dominance of
gastropods negatively affected species diversity by monopolizing space and nutritional
resources, therefore reducing the settlement of other vent species. The grazing of new
recruits may also limit species diversity. Successional community dynamics leading to the
development of tubeworm assemblages may thus result in the diversification of the habitats
and of the species therein, and by a complexification of the trophic network, as suggested by
Sarrazin *et al*. (2002) (Sarrazin et al., 2002).

**4.2. Trophic structure of tubeworm assemblages**
The *R. piscesae* tubeworm assemblages of the Grotto hydrothermal edifice harbour a
relatively diverse heterotrophic fauna. The isotopic analyses conducted on the most
dominant vent species within the bushes revealed a high degree of resemblance in trophic
structure among the six faunal assemblages.

Hydrothermal food webs are generally based on two main energetic pathways: the transfer
of energy from symbionts to host invertebrates and the consumption of free-living microbial
production (Bergquist et al., 2007). In the present study, the contrasting isotope
compositions of the gastropods *P. variabilis*, *L. fucensis* and the polychaete *R. piscesae*
suggest three large pools of isotopically distinct, symbiotic and/or free-living microbial
production available to primary consumers. The high $\delta^{13}$C values of *R. piscesae* were



associated with chemosynthetic endosymbiosis linked to thiotrophic symbionts (Hügler and
Sievert, 2011). *R. piscesae* contributed to 86 % of the assemblage biomass, but few species
displayed similar $\delta^{13}$C values, suggesting that species deriving their food sources from
siboglinid tubeworms are rare. Similar observations, where engineer species contribute to
the community more as a habitat than as a food source, have been reported in *R. piscesae*
tubeworm bushes from the Easter Island vent site (Bergquist et al., 2007) or in
*Bathymodiolus azoricus* mussel bed communities on the Tour Eiffel hydrothermal edifice
(Lucky Strike, Mid-Atlantic Ridge) (De Busserolles et al., 2009). The low degree of
exploitation of this large biomass and potential food resource suggests that *R. piscesae* plays
a primarily structuring role in vent ecosystems rather than a trophic role. Nevertheless, the
$\delta^{13}$C and $\delta^{15}$N values of polynoid predators *Branchinotogluma tunnicliffeae* and
*Lepidonotopodium piscesae* were consistent with a diet including *R. piscesae* tubeworms.
Predation on tubeworms was confirmed by a video sequence from the ecological
observatory module TEMPO-mini, deployed on the Grotto hydrothermal edifice (ONC
observatory; Video S1). The $^{13}$C-depleted stable isotope compositions of *P. variabilis* suggest
a possible symbiosis with chemoautotrophic bacteria or reliance on feeding on a very
specific free-living microbial community that depends on a $^{13}$C-depleted carbon source
(Bergquist et al., 2007). To date, no study has reported the presence of chemoautotrophic
symbionts in *P. variabilis*, but symbioses have been described for other species from the
Provannidae family (Windoffer and Giere, 1997). With an intermediate $\delta^{13}$C composition
between *R. piscesae* and *P. variabilis*, *L. fucensis* gastropods seem to represent a major
energetic pathway in these vent communities. In addition, the different food webs obtained
in this study revealed that most vent species display an isotope composition centred on *L.*
*fucensis*. The position of *L. fucensis* at the base of the food web probably reflects direct
access to suspended food particles from hydrothermal fluid emissions. The high densities
and large biomass of *L. fucensis* in tubeworm bushes, and its capacity to exploit different
food sources through different feeding modes (Bates, 2007), may exert a high pressure on
the availability of nutritional resources and, therefore, lead to an important role in
structuring vent communities. Whenever present, the *Paralvinella* species, which are non-
selective deposit-feeders, also displayed low $\delta^{15}$N values, suggesting a role at the base of the
food web. The stable isotope composition of *Paralvinella* species was much more variable





among samples than for the former three species, suggesting a possible variability in
nutrient sources.

Like in many vent food webs (Van Dover and Fry, 1994; Levesque et al., 2005; Limén et al.,
2007), Grotto primary consumers were dominated by grazers and deposit feeders. The high
diversity, densities and biomass of bacterivores emphasize the importance of free-living
bacteria in the establishment and maintenance of the structure of the vent food web
(Bergquist et al., 2007). This guild was mainly represented by the gastropods *P. variabilis*, *D.*
*globulus* and *L. fucensis* and by the polychaetes *P. sulfincola*, *P. palmiformis*, *P. pandorae* and
*Paralvinella dela*. The polychaete *P. sulfincola* can feed directly on microbial biofilms on the
substratum around its tube opening (Grelon et al., 2006), which may explain the low $\delta^{15}$N
values of alvinellids in the present study. Like *Paralvinella grasslei* and *Paralvinella*
*bactericola* at vent sites of the Guaymas Basin (Portail et al., 2016), the alvinellid species
found at Grotto had comparable $\delta^{13}$C values but different $\delta^{15}$N signatures. The species *P.*
*pandorae* showed a depleted $\delta^{15}$N signature relative to other alvinellid species. A previous
study of spatial isotope variability among three sympatric alvinellid species, *P. palmiformis*,
*P. sulfincola* and *P. pandorae* on the JdFR reported that this difference in $\delta^{15}$N isotope
composition was closely related to food-source partitioning and/or to spatial segregation
(Levesque et al., 2003). The comparatively small size of *P. pandorae* (Lelièvre Y., personal
observation) compared with other alvinellid species may be the result of interspecific
competition for food resources and/or a diet based on an isotopically distinct microbial
source. The wide range of $\delta^{13}$C signatures in bacterivores, coupled with the high interspecific
variability in the isotopic space, suggest a large, diversified microbial pool in the
hydrothermal ecosystem and high variability in isotope ratios in dominant microbial taxa.
Detritivore/scavenger species were observed at an intermediate trophic level, between the
bacterivore and predator feeding guilds. This guild was represented by a low number of
species including the gastropod *B. thermophilum*, the ampharetid *A. carldarei* and the
orbiniid *Berkeleyia sp. nov*. The predator-feeding guild was represented by the highest $\delta^{15}$N
values. High predator diversity was found in our vent assemblages, and was associated with
a wide range of $\delta^{13}$C values, covering the isotopic spectrum of lower trophic level consumers
(i.e. bacterivores as well as scavengers/detritivores). This guild of predators appears to be
dominated by polychaetes, which tend to show the highest $\delta^{15}$N values. Whenever present,





the syllid *Sphaerosyllis ridgensis*, the polynoid *Levensteiniella kincaidi* and the hesionid
*Hesiospina sp. nov*. displayed the highest $\delta^{15}$N values, suggesting that they play the role of
top predators in the benthic food web. Similarly, the solenogaster *Helicoradomenia juani*
consistently displayed higher $\delta^{15}$N values than other molluscs, indicating a predator trophic
position. Except for the polynoid *L. kincaidi*, whose isotopic variability seemed to reveal a
nutrition based on highly diversified food resources, stable isotope analyses conducted on
predators revealed narrow ranges of $\delta^{13}$C and $\delta^{15}$N values at the species scale, suggesting
the dominance of specialist-feeding strategies, as was the case for the bacterivores. An
accurate assessment of food sources and a description of the meiofaunal communities
would be necessary to better understand the functioning of these chemosynthetic
communities and their trophic structures.

**4.3. Ecological niche partitioning**
Vent species on the Grotto hydrothermal edifice exhibit high isotopic heterogeneity that
reflects the complexity of vent ecological networks. The distribution of species in the bi-
dimensional isotopic space depends on their diets, environmental conditions and biotic
interactions, which together define the concept of species ecological niche (Newsome et al.,
2007) or the realized species trophic niche (Bearhop et al., 2004). Here, the fact that most of
the isotopic space was occupied by isotopically distinct species shows that the available food
resources are partitioned within the community. Although the $\delta^{15}$N variability among
primary consumers did hinder our inference of trophic levels based on nitrogen isotopes,
these communities are unlikely to host more than three trophic levels, given the overall $\delta^{15}$N
ranges. Moreover, although predators were quite diverse, they only represented a minor
part of the biomass, suggesting that Grotto vent communities are mostly driven by bottom-
up processes. Food webs of chemosynthetic ecosystems – such as hydrothermal vents and
cold seeps – do not appear to be structured along predator-prey relationships, but rather
through weak trophic relationships among co-occurring species (Levesque et al., 2006;
Portail et al., 2016). Habitat and/or trophic partitioning are important structuring processes
at the community scale (Levesque et al., 2003; Levin et al., 2013; Portail et al., 2016). Our
results corroborate with those from Axial Volcano in the JdFR (Levesque et al., 2006) and the
Guaymas basin (Portail et al., 2016), where habitat heterogeneity induces spatial partitioning
of trophic niches, leading to a spatial segregation of species and species coexistence





(Levesque et al., 2006). Although the observed isotope variability (standard deviations) in
Grotto vent species suggests the occurrence of both trophic specialists and generalists
within the assemblages, the majority of vent species exhibited low standard deviations,
suggesting a predominantly specialist feeding behaviour. As already shown in previous
studies of vent sites with alvinellids (Levesque et al., 2003) and sulfidic sediments at
methane seeps with dorvilleid polychaetes (Levin et al., 2013) in the north-east Pacific, food
partitioning may occur between different species of the same or closely related taxonomic
family, allowing species coexistence through occupation of distinct trophic niches. For
example, hydrothermal vent gastropods were numerically dominant in all *R. piscesae* bushes
collected on the Grotto edifice and their isotope compositions were fairly diverse.
Gastropods exhibit great diversity in feeding strategies, and as a result they are found in a
wide variety of niches where they exploit many food sources (Bates et al., 2005; Bates,
2007). The isotope composition of *P. variabilis* indicated low $\delta^{13}$C and $\delta^{15}$N values. *L. fucensis*
gastropods had higher $\delta^{13}$C and $\delta^{15}$N values than *P. variabilis* but a similar range of $\delta^{13}$C as
*Clypeosectus curvus* and *D. globulus*. However, these latter two species occupy an upper
position in the trophic structure of their communities. The great ecological success of *L.*
*fucensis* in vent habitats may be attributed to a combination of several characteristics. First,
this species is characterized by a broad trophic plasticity that includes: (*i*) grazing on
siboglinid tubeworms and hard substrata (Fretter, 1988), (*ii*) active suspension feeding
(Bates, 2007) and (*iii*) harbouring filamentous bacterial epibionts in its gills, which – via
endocytosis – may contribute to the animal's nutritional requirements (Bates, 2007; Fox et
al., 2002). In addition, the early maturity, high fecundity, and continuous gamete production
of *L. fucensis* may help to maintain the large populations on the edifice (Kelly and Metaxas,
2007). Stacking behaviour near fluid emissions also suggests that *L. fucensis* is an important
competitor for space and food in the community (Tsurumi and Tunnicliffe, 2003). *L. elevatus*,
the ecological equivalent of *L. fucensis* on the East Pacific Rise, is a prey for the vent zoarcid
fish *Thermarces cerberus*; the reduced limpet population promotes the successful
settlement and growth of sessile benthic invertebrates such as tubeworms (Micheli et al.,
2002; Sancho et al., 2005). The potential absence of an equivalent predator for *L. fucensis*
and the biological characteristics detailed above may explain its ecological success on the
north-east Pacific vent sites. In contrast, the nutrition of *D. globulus* is based on the grazing
of organic matter only (Warén and Bouchet, 1989). However, its small size allows it to



exploit interstitial spaces that are not available to larger fauna (Bates et al., 2005). Finally, *P.*
*variabilis* was relatively less abundant than the other two species, but appeared to exploit a
different thermal niche than *L. fucensis* and *D. globulus* (Bates et al., 2005). On the other
hand, the isotope composition of *B. thermophilum* clearly differentiates that species from
the other gastropods with higher $\delta^{13}C$ signatures. Differences in the diets of co-occurring
species may contribute to the high abundance – such as *L. fucensis* and *D. globulus* – and
diversity of vent gastropods through niche partitioning (Govenar et al., 2015).

Habitat specialization among co-occurring vent species may drive differences in their diets
(Govenar et al., 2015), facilitating species coexistence in heterogeneous habitats such as
hydrothermal ecosystems. We hypothesized that in vent engineering ecosystems, food webs
display a spatial structure at small scale with regard to the microhabitats generated by the
3D architecture of biogenic structures that promote high interspecific trophic segregation.
The spatial segregation of trophic niches by environmental gradients limits the occurrence of
biotic interactions such as predation and competition for resources between species sharing
a common spatial niche (Levesque et al., 2006). Vent food webs may therefore be structured
through the interplay between the availability and diversity of food sources and the abiotic
and biotic conditions structuring species distribution.

**5. Conclusion**
This study provides the first characterization of the macrofaunal diversity and trophic
ecology of vent communities associated with *R. piscesae* tubeworm assemblages on the
Grotto hydrothermal edifice. Like many vent structures (Cuvelier et al., 2011; Sarrazin et al.,
1997), the Grotto hydrothermal edifice is inhabited by a mosaic of habitats and faunal
assemblages that may represent different successional stages characterized by different
abiotic conditions. Our results show that the development of *R. piscesae* tubeworms
introduces complexity and heterogeneity in the hydrothermal environments and exerts a
strong influence on ecosystem properties. The 3D structure of these tubeworms enhances
community diversity and thereby increases the potential trophic interactions between vent
species in the food web. Environmental gradients provided by the interstitial spacing of
intertwined tubeworms generate a multitude of ecological niches and contribute to the
partitioning of nutritional resources, leading to the species coexistence. Habitat



modifications incurred by *R. piscesae* bushes may thus directly stimulate the development of
complex food webs. Thorough knowledge of hydrothermal biodiversity and ecological
functioning of these remote ecosystems is necessary to determine their uniqueness and
contribute to the protection and conservation of this natural heritage.

**Author's contributions**

M.M., J.S. and P.L. designed and supervised the research project. Y.L., J.M., T.D. and S.H.:
data acquisition and analyses. Y.L., M.M., J.S., G.S. and P.L. conceived the ideas and
contributed to the interpretation of the results. All authors contributed to the writing
process and revised the manuscript.

**Competing interests**

The authors declare that they have no conflict of interest.

**Acknowledgments**

The authors thank the captains and crews of the R/V *Thomas G. Thompson* and E/V *Nautilus*,
the staffs of Ocean Networks Canada and ROV *Jason* and *Hercules* pilots during the "*Ocean*
*Networks Canada Wiring the Abyss*" cruises in 2015 and 2016. We thank also Kim Juniper
and the government of Canada for work permits to study in Canadian waters (XR281, 2015;
XR267, 2016). Thanks also to Pauline Chauvet for faunal sampling during the ONC 2016
cruise. This research was supported by a NSERC research grant to P.L. and IFREMER funds. It
was also funded by the *Laboratoire d'Excellence* LabexMER (ANR-10-LABX-19) and co-funded
by a grant from the French government under the *Investissements d'Avenir* programme. We
are also grateful to the numerous taxonomists around the world who contributed to species
identification and to the laboratory *Centre de Recherche sur les Interactions Bassins Versants*
*- Écosystèmes Aquatiques* (RIVE) at the Université du Québec à Trois-Rivières (Canada) for
the isotope sample processing. The manuscript was professionally edited by Carolyn Engel-
Gautier. This paper is part of the Ph.D. thesis of Y.L. carried out under joint supervision
between Université de Montréal and Université de Bretagne Occidentale/Ifremer.





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





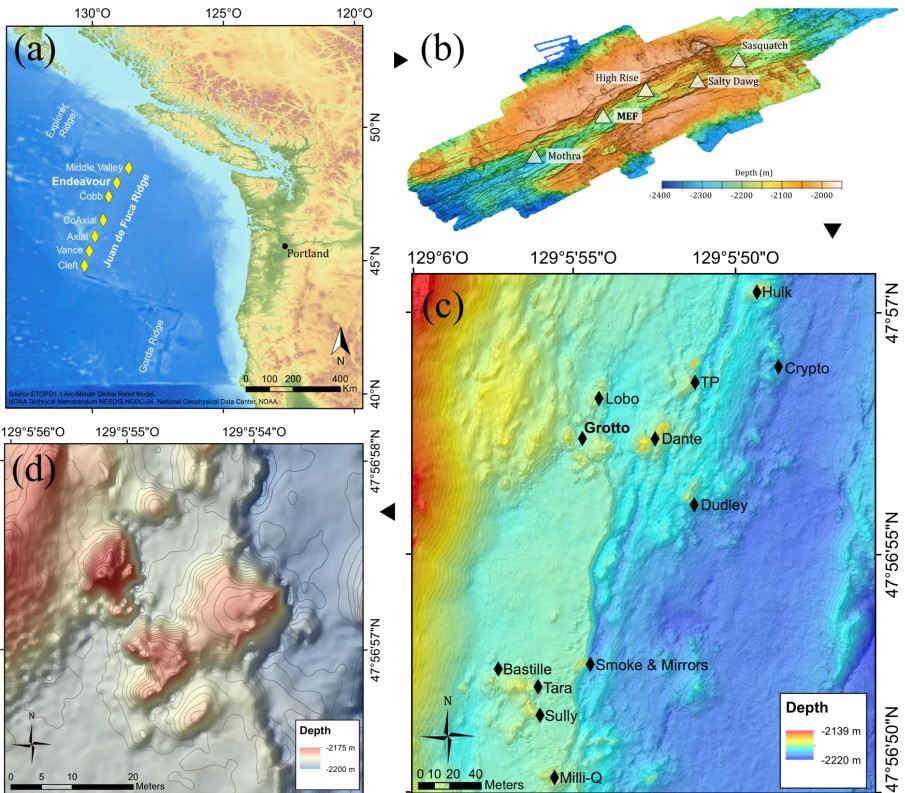

**Figure 1.** (a) Location of the Juan de Fuca Ridge system and the seven segments (yellow diamonds). (b) High-resolution bathymetric map of the Endeavour Segment, with the locations of the five main active vent fields (white triangle). (c) Location map of the Main Endeavour vent field indicating the positions of hydrothermal vent edifices (black diamonds). (d) Bathymetric map of the Grotto active hydrothermal edifice (47°56.958'N, 129°5.899'W). The 10 m high sulfide structure is located in the Main Endeavour vent field.










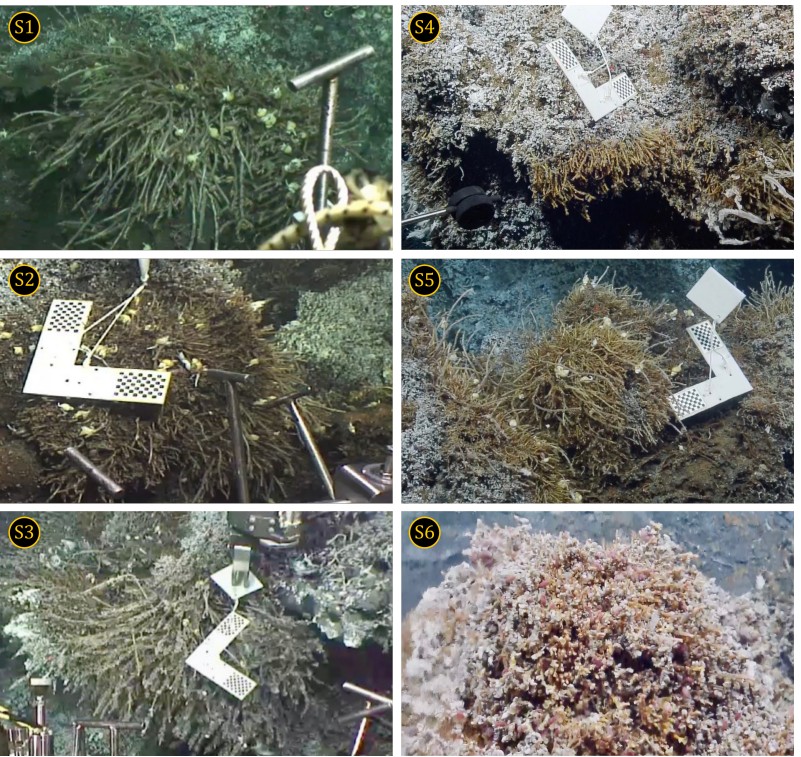

**Figure 2.** Hydrothermal communities collected on the Grotto edifice (Main Endeavour, Juan de Fuca Ridge) during *Ocean Networks Canada* oceanographic cruises *Wiring the Abyss 2015 and 2016*.


















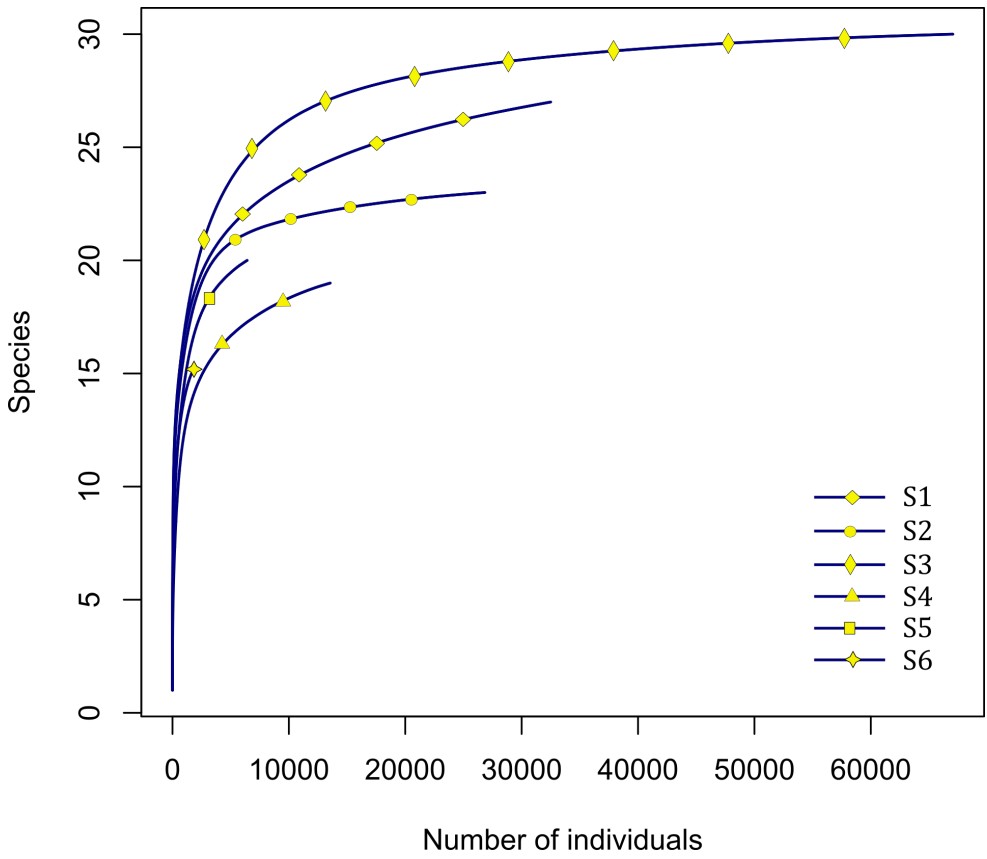

**Figure 3.** Rarefaction curves for species richness in six vent assemblages (S1 to S6) sampled on the Grotto hydrothermal edifice.














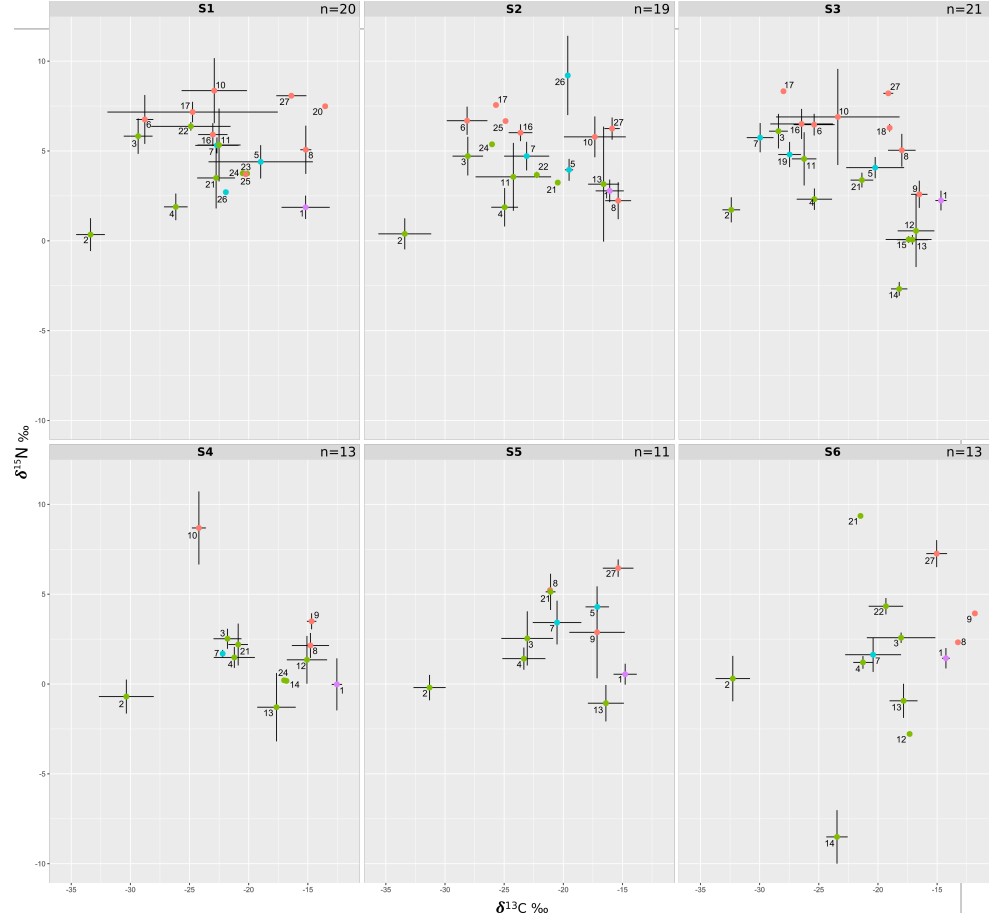

**Figure 4.** Stable isotope bi-plots showing vent consumers' isotope signatures (mean $\delta^{13}$C versus $\delta^{15}$N values ± standard deviation) for the six vent assemblages sampled on the Grotto hydrothermal edifice. Each vent species is designated by a number: 1 = *Ridgeia piscesae*; 2 = *Provanna variabilis*; 3 = *Depressigyra globulus*; 4 = *Lepetodrilus fucensis*; 5 = *Buccinum thermophilum*; 6 = *Clypeosectus curvus*; 7 = *Amphisamytha carldarei*; 8 = *Branchinotogluma tunnicliffeae*; 9 = *Lepidonotopodium piscesae*; 10 = *Levensteiniella kincaidi*; 11 = *Nicomache venticola*; 12 = *Paralvinella sulfincola*; 13 = *Paralvinella palmiformis*; 14 = *Paralvinella pandorae*; 15 = *Paralvinella dela*; 16 = *Hesiospina sp. nov.*; 17 = *Sphaerosyllis ridgensis*; 18 = *Ophryotrocha globopalpata*; 19 = *Berkeleyia sp. nov.*; 20 = *Protomystides verenae*; 21 = *Sericosura* sp.; 22 = *Euphilomedes climax*; 23 = *Xylocythere sp. nov.*; 24 = Copepoda; 25 = *Copidognathus papillatus*; 26 = *Paralicella vaporalis*; 27 = *Helicoradomenia juani*. Known trophic guilds are distinguished by a colour code: pink: symbiont; green: bacterivores; blue: scavengers/detritivores; red: predators. For more information on the interpretation of guilds, please consult the web version of this paper.




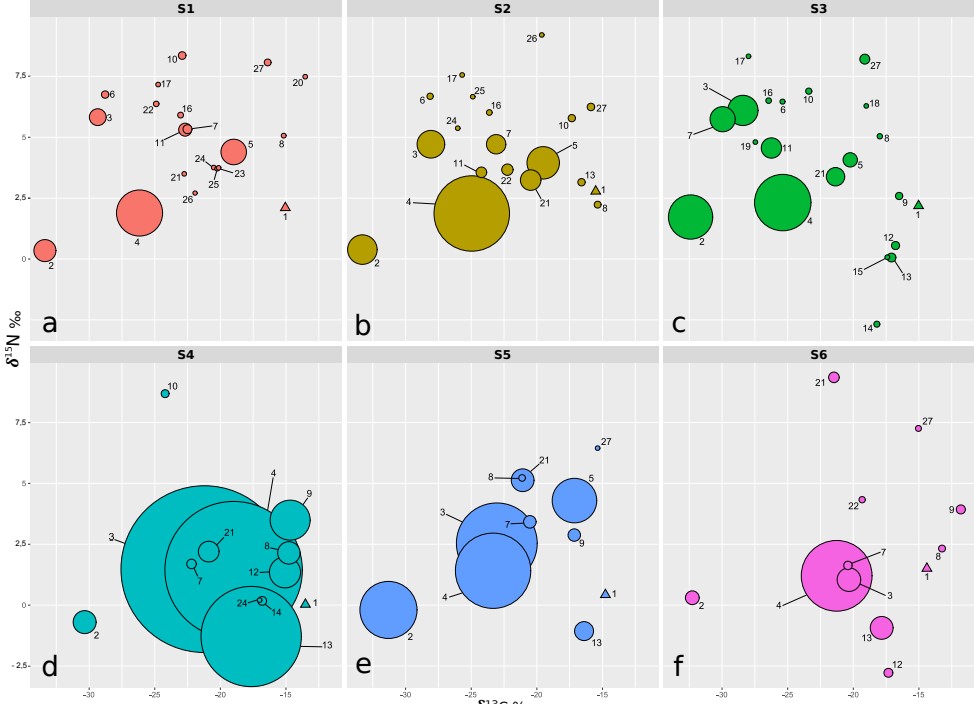

**Figure 5.** Stable isotope bi-plots showing vent consumers' isotope signatures weighted by biomass per cubic meter (filled circles) for the six vent assemblages (S1 to S6) sampled on the Grotto hydrothermal edifice. Considered as a habitat, the biomass of *Ridgeia piscesae* (denoted by a triangle symbol) is not shown. Each vent species is designated by a number: 1 = *Ridgeia piscesae*; 2 = *Provanna variabilis*; 3 = *Depressigyra globulus*; 4 = *Lepetodrilus fucensis*; 5 = *Buccinum thermophilum*; 6 = *Clypeosectus curvus*; 7 = *Amphisamytha carldarei*; 8 = *Branchinotogluma tunnicliffeae*; 9 = *Lepidonotopodium piscesae*; 10 = *Levensteiniella kincaidi*; 11 = *Nicomache venticola*; 12 = *Paralvinella sulfincola*; 13 = *Paralvinella palmiformis*; 14 = *Paralvinella pandorae*; 15 = *Paralvinella dela*; 16 = *Hesiospina sp. nov.*; 17 = *Sphaerosyllis ridgensis*; 18 = *Ophryotrocha globopalpata*; 19 = *Berkeleyia sp. nov.*; 20 = *Protomystides verenae*; 21 = *Sericosura* sp.; 22 = *Euphilomedes climax*; 23 = *Xylocythere sp. nov.*; 24 = Copepoda; 25 = *Copidognathus papillatus*; 26 = *Paralicella vaporalis*; 27 = *Helicoradomenia juani*. For legibility, the biomass of *P. pandorae* in collection S6 is not shown.
