# Peer review of "tubeworm assemblages on the Juan de Fuca Ridge"

_Biogeosciences, 2017_

## Referee Comment (RC1) · R. Boschen (Referee) · 27 Oct 2017

Review of bg-2017-411

General comments

Overall the manuscript is well written and the concepts discussed are interesting. The approach is not novel; papers have been written on the community structure and stable isotope characters of these species previously. However, the extension of this work to the concept of a 'trophic network' through linkages to biomass has not been done in this location previously and the perspectives it provides are interesting. Some areas of the manuscript require clarification and there are questions about some aspects of the approach. I couldn't find the tables referenced in the manuscript: some of my concerns may be addressed by being able to evaluate these.

Specific comments

- Species list and abundances don't seem to be provided, although they are mentioned as being in Table 2. These should be made available, at least in supplementary format. Species list provision is particularly important for this study as all the samples were collected within the Endeavour Hydrothermal Vents Marine Protected Area (EHV MPA); species occurrence data would be very useful to the managing authority Department of Fisheries and Oceans (DFO). No mention of EHV MPA is made within the manuscript.
- "36 macrofaunal taxa were identified to the species level" (line 33); "35 macrofaunal taxonomic groups" (line 287 & 367): which is it - 36 species, or 35 taxa? Please clarify.
- Line 37: "fairly long tubes". Please define, how long?
- Lines 98 – 101: limited behavioural observations and gut contents analyses, compared to wide use of stable isotopes. For JdFR vents, there are many observations on behaviour, especially feeding, and some on gut contents, in the original species descriptions. True, there are few studies specifically addressing feeding behaviour or gut contents, compared to dedicated stable isotope studies. However, stable isotopes are just one of many tools to evaluate feeding, and like other techniques, stable isotopes have their limitations. It would be helpful to have a few sentences acknowledging the pros & cons of stable isotopes compared to other techniques available to assess feeding and trophic structure.
- Line 115: "despite the relatively low diversity of the deep-sea community" - do you mean low diversity of the hydrothermal community? Lines 63-64: "[hydrothermal vent communities have] low species diversity compared with adjacent deep-sea and coastal benthic communities". Please clarify.
- Line 122: Urcuyo *et al*. 2003. This is not the first reference to *Ridgeia piscesae* populations along the JdFR; please consider replacing or supplementing with an earlier reference. For example, the Revision of Ridgeia was published by Southward *et al.* in 1995; records of Ridgeia along the JdFR will extend to before this reference. "Southward *et al.* (1995) Revision of the species of *Ridgeia* from northeast Pacific hydrothermal vents, with a redescription of *Ridgeia piscesae* Jones (Pogonophora: Obturata = Vestimentifera. *Canadian Journal of Zoology*. 73: 282-295.
- Line 153: Lelievre *et al.* 2017. There is an earlier study on tidal influence on *R. piscesae* that was not referenced in Lelievre *et al.* 2017 and should probably be referenced here: Johnson & Tunnicliffe (1985) Time-series measurements of hydrothermal activity on northern Juan de Fuca Ridge. *Geophysical Research Letters* 12: 10, 685-688.

- Line 177: "intense". Intense in terms of what? This could probably be removed so the sentence reads "The MEF… is the most active of the five hydrothermal fields…"
- Line 183: "Like many other MEF hydrothermal edifices…" This only has one reference, Sarrazin *et al.* 1997. This reference only characterises one hydrothermal structure, S & M. You need more references here to support the 'many' in your sentence; Govenar *et al.* 2002, Bergquist *et al*. 2007 and Urcuyo *et al.* 2007 are all possible additions. The latter is not in your reference list: Urcuyo *et al.* (2007) Growth and longevity of the tubeworm *Ridgeia piscesae* in the variable diffuse flow habitats of the Juan de Fuca Ridge. *Marine Ecology Progress Series* 344: 143-157.
- Lines 203-206: macrofauna and meiofauna sampling. If macrofaunal specimens are ">250 um" and meiofauna are "<63 um" then what are the fauna that are <250 um but > 63 um? The definition of macrofauna is variable, but 250 um is on the smaller end. 0.5 mm or 1 mm sieves are more common for macrofauna retention. Meiofauna would usually be considered to be 63 um – 0.5 mm: <63 um would be microfauna.
- Lines 209-212: trophic guilds. If trophic guilds were assigned to species prior to your stable isotope analysis, there needs to be a table detailing which species were assigned to different guilds, preferably with the reference(s) used for the designation and an indication if species level of higher was available. From the way this is written in the methods, it almost sounds like a circular argument: literature was used to assign trophic guilds (which will mostly contain stable isotope information) and then stable isotopes were used to assess the validity of these groups. There needs to be more detail here as to how the 'trophic guild' information was obtained and how it was applied.
- Line 215-222: density of individuals per m-2. This is presumably based on the surface area of the sulphide chimney the tubeworms were attached to. But it is not really the surface area for the fauna – that would be the tubeworm surface area. I would be very cautious about using ind. $m^{-2}$ based on the chimney area sampled. Equally, as the basis for ind. $m^{-3}$ I would be cautious. I appreciate this volume takes into consideration average tubeworm length, but it doesn't take any account the width of the tubes or how twisted and intermeshed they are, which all contribute to the habitat area the tubeworms provide. Tsurumi and Tunnifliffe (2001), Tsurumi and Tunnicliffe (2003) & Marcus and Tunnicliffe (2009) all use tubeworm area standardised abundances for this reason.
- Lines 223-226: biomass estimates. I find these lines a little unclear… for the total dry mass of each species, did you dry all individuals of those species? Did you just dry the 3 – 10 individuals that you had randomly selected? If just the selected individuals, were these dried separately or together? What do you mean by "absolute biomass of each species was calculated by multiplying the relative biomass by the abundance of each species"? Did you take the mean weight of your randomly selected individuals and multiply this by your abundance? Please provide some clarification on the biomass estimates section.
- Lines 230-231: gut content. I'm not clear what you mean here by "in the case of intermediate-sized specimens, the gut content was removed". Does this mean you did the stable isotope analysis on the whole individual, minus its gut? Or that you did the stable isotope work on the gut contents? Please clarify.
- Lines 265-266: sample volume as a proxy for habitat complexity. I am not entirely convinced by this, the long tubeworms seen at very low flow sites would provide a larger 'volume' by your definition, but their straight stiff tubes do not interlock in the same complex structure

that intermediate flow tubes or even some high flow tubes do. I.e., shorter tubes may make for a smaller volume but it doesn't necessarily make for reduced habitat complexity. Southward and Tunnicliffe 1995, Tsurumi and Tunnicliffe (2003) & Tunnicliffe *et al*. 2014 discuss the phenotypic complexity of *R. piscesae* in terms of tube structure and the implications this has for faunal communities.

- Line 293: densities in $m^{-3}$. These numbers are incredibly large and speak to my point about tubeworm surface area… what does the number of individuals per $m^{-3}$ really mean for benthic organisms? Most of these species will be attached to substrata, i.e. the tubeworms, so that tubeworm surface area, measured in $cm^{-2}$ (as in Tsurumi and Tunnicliffe 2001) would probably be more relevant on a habitat scale than $m^{-3}$.

- Lines 286-299: macrofaunal densities. I presume these densities exclude *R. piscesae* abundance?

- Line 315: acarida. *Copidognathus papillatus* is in the class 'Arachnida', subclass 'Acari'…. Family 'Halacaridae'. I'm not sure where 'acarida' comes from but it is not an accepted taxonomic name in the World Register of Marine Species (WORMS): http://www.marinespecies.org

- Lines 350-351: "With a biomass ranging from…" In this and subsequent sentences, I presume you are referring to the biomass ranges across all samples at Grotto? Could you please clarify this in the text?

- Lines 361-362: feeding guilds. See previous comment on how trophic guilds were assigned and applied. I am unclear here whether the discussion of biomass and guild relationships refers to guilds applied to taxa using the literature (including previous stable isotope studies) or whether the authors have re-assigned guilds from their own stable isotope data. A table with the species and guilds assigned using the literature, with an additional column indicating whether the present study agrees with these designations would be helpful.

- Line 390: "a pattern that has also…." I presume here you mean the pattern of dominance by a few species? This is worth clarifying, as it could be misread that the dominance of *L. funcensis, D.globulus, A carldarei* specifically is the pattern you are referring to. Consider breaking it into two sentences: "*R. piscesae* tubeworm assemblages sampled on the Grotto edifice were characterized by the dominance of a few species, such as *L. fucensis, D. globulus* and *A. carldarei.* Numerical dominance by a few species is a pattern that has also been observed from other hydrothermal sites on the world oceans…."

- Lines 400-401: sample size. How much smaller were the samples from 2016, in terms of surface area/volume? How much material (as an estimate) was lost during collection?

- Line 412: what do you mean by "a high level of complexity?" a complex tubeworm bush structure? This needs some clarification.

- Lines 413-416. With the limited information you have on the character of your tubeworms from each sample, I'm not convinced you have sufficient evidence for this statement. You haven't assessed the nature of the biogenic structure for your samples (i.e. the physical structure of your tubeworm bushes); you've calculated the potential volume of seawater your tubeworm bushes could occupy, which is not the same.

- Lines 429-430: why do your density ranges for *A. carldarei* exclude S4 and S6? If you're talking about the Grotto edifice as a whole, shouldn't these two samples be included?

- Line 430: High densities of what? *A. carldarei*? Please clarify in the text.

- Lines 430-431: "…may be related to the specificity of this family with high ecological tolerance and, therefore…" Specificity and tolerance are opposite ends of a spectrum, consider rephrasing to "….may be related to this ampharetid's tolerance to environmental conditions and, therefore…."

- Lines 441-442: "The grazing of new recruits may also limit species diversity" Do you mean the grazing action of new recruits to the community limits species diversity, or the grazing action of the community on new recruits limits species diversity? Please clarify.

- Lines 455-457: three pools of isotopically distinct microbial production. Do you think *L. fucensis* really feeds on its own separate pool of microbial production? Or does its contrasting isotopic signature reflect the fact that it may be utilising multiple microbial sources at any one time? As you mention later (lines 569-575), it can host bacteria on its gill filaments and eat them, along with grazing and suspension feeding activities. It might be worth introducing this concept here instead of the brief reference to Bates (2007) at line 485.

- Lines 461-462: "…species deriving their food sources from siboglinid tubeworms are rare." I don't quite agree here – there are multiple species that do eat *R. piscesae*, including *B tunnicliffae* as you mention at line 469. *Buccinum thermophilum* also preys on *R. piscesae*. That the isotopic signatures don't closely match these predators to their prey doesn't mean that the predators are rare; instead it suggests the predators are not specialist, i.e. they eat more than just *R. piscesae*.

- Line 471: "Predation on tubeworms was confirmed…" Predation by whom? There are many images/video clips of *B. tunnicliffae* predating on *R. piscesae*; do you have imagery of *L. piscesae* doing the same? This sentence needs some clarification.

- Lines 480-482: "most vent species display an isotopic composition centred on *L. fucensis*" What do you mean by this? That most of the species have an isotopic composition similar to *L. fucensis*? Please clarify.

- Lines 483-487: the role of *L. fucensis.* Do you think that the wide-ranging feeding strategies of this limpet really exert pressure on the availability of resources for others? Or does the range of strategies open up more opportunities for *L. fucensis*, allowing them to take advantage of resources that others cannot? See you text at lines 569-575.

- Lines 487-490: *Paralvinella* species. These line are somewhat unclear, are all *Paralvinella* selective deposit feeders? Was the stable isotope composition more variable in terms of intraspecific variation or in terms of interspecific variation among samples?

- Line 494: "Grotto primary consumers were dominated by grazers and deposit feeders" Where is your evidence/data for this? A table of your trophic guild designations would be really useful to refer to here.

- Lines 500-501: *P. sulficola* low nitrogen values. This sentence needs some clarification; the feeding behaviour of *P sulfincola* can only explain the low nitrogen values for that species, not all *Paralvinella* species, as they may (and in fact do) display different feeding strategies.

- Line 508: "the comparatively small size of *P. pandorae*" How large are the *P. pandorae* in your samples compared to the size range of your other *Paralvinella* species?

- Lines 517-520: wide range of carbon values. Does this mean the carbon values provide less supporting evidence for your designated guilds than the nitrogen values? As phrased, it appears that you are choosing to go with the nitrogen data as it better fits your guild

concepts, despite the range in carbon... Please could you clarify? Again, having a table with your guild designations for each species, whether the carbon and nitrogen values from your study support these, and even how your values compare to other studies (e.g. Bergquist *et al*. 2007) would be really helpful. See Table 1 in Bergquist *et al.* 2007 for a starting point.

- Lines 521-529: These lines are unclear with respect to *L. kincaidi*. The high nitrogen values make it a "top predator", but isotopic variability suggests it has "highly diversified food resources". So it isn't just a predator but also exhibits other feeding strategies? Please provide some clarification here, again a table of isotopic values by species and your guild designations (*cf* Bergquist *et al.* 2007) would be really useful for interpretation.

- Lines 539-541: How did your results overall compare to Bergquist *et al.* 2007? Did you get similar values and ranges for the same species? It would be helpful to compare your work to theirs here.

- Lines 561-566: gastropods. The group gastropoda is very large and the species you have; *D. globulus, P. variabilis* & *L. fucensis* come from completely different families, they are even in different sub-classes (Neomphalina, Caenogastropoda, Vetigastropoda respectively) so I'm not convinced the concept of "closely related" (line 560) in relation to these species and resource portioning is reasonable. They are not closely related; it is not so surprising that they have different feeding strategies and occupy different niches.

- Lines 588-589: *B. thermophilum* carbon signatures. This species has been observed predating on tube worms, could the higher carbon values support this and/or other predatory behaviours?

- Lines 596-598 & 611-613: I still have concerns that your characterisation of the tubeworm habitat is not sufficient to make this statement, see previous comments relating to tubeworm plasticity and habitat volume as a proxy.

Technical corrections

- Line 35 & 36: would read better if gastropods are listed first. See previous sentence.
- Lines 88 – 98: multiple occurrences of 'e.g.' and 'etc.'; could these sentences be more specific?
- Line 172: Middle Valley isn't really a site *per se*; it's more of a seabed area and there are sites within it. Might just read better as 'Middle Valley' without 'site'.
- Lines 174-175: would read better as "The five major vent fields – Sasquatch, Salty Dawg, High Rise, Main Endeavour Field (MEF) and Mothra – found along the Endeavour axial valley are each separated by 2-3 km." Please note that MEF stands for Main Endeavour Field, not just Main Endeavour.
- Line 183: "a relative stability in years": would read better as "but is relatively stable across years"
- Line 234: "About 1.3-1.4 mg of the powder was precisely measured" consider substituting 'about' for 'approximately'.
- Line 268: Table 1. Where is Table 1? I can't find it in the pdf. I also cannot find Table 2!
- Lines 279-280: the lowest diversity values. This sentence would read better if the lowest diversity values were listed first, in the order of increasing diversity values.
- Lines 301, 329 & 337: "more specifically". You might consider an alternative start to these sentences, such as "In more detail".

- Line 323: "…S6 was also dominated *L. fucensis*" should read "…S6 was ably dominated by *L. fucensis*"
- Line 384: "engineer polychaete" should probably read "ecosystem engineering polychaete"
- Lines 372-376. Consider rephrasing slightly to make it clear the reference is for all the species richness records. Alternative phrasing could put the reference at the start of the sentence: "Tsurumi and Tunnicliffe (2003) reported the presence of 39 macrofaunal species…. 19 species in 2 collections from the CoAxial Segment (JdFR)."
- Line 379: "engineer mussel beds: should probably read "ecosystem engineering mussel beds"
- Line 382: "worldwide hydrothermal ecosystems". In the context of this sentence, I think you mean biogeographic regions or provinces? Consider changing to "Faunal dissimilarities between biogeographic regions…"
- Lines 442-445: this sentence is to me unclear. Please consider re-phrasing/clarifying. Also note repeat Sarrazin *et al.* (2002) reference.

---

## Referee Comment (RC2) · Anonymous Referee #2 · 26 Jan 2018

The manuscript by Lelièvre and colleagues presents interesting information on the structure, biodiversity, and trophic structure of hydrothermal vent sites on the Main Endeavor Field on the Juan De Fuca ridge. This manuscript and their data take a nice holistic approach and compare a variety of different communities that they suggest are different successional stages based on observations there, thus link successional patterns to the greater community and trophic structure that live there. The manuscript is well crafted and they found that predator-prey relationships were not as dominant as the important role of ecosystem engineers in structuring the communities. The research is also important as it provides a nice baseline for future studies that work at these locations, which are near a cabled array system so this is likely to be highly referenced site in the future. My only complaint is a slightly cursory treatment of the isotopic data and the use of individuals per mˆ3 instead of mˆ2 for benthic communities which I believe are muddling some of the results. This is a nice manuscript that advances the field.

One aspect that I think that could improve the paper is a hypothetical diagram that shows how the communities change over time (as suggested that the communities are different temporal stages), the fauna and diversity associated with those stages and the trophic structure as a result. This is a suggestion only, but it would be a wonderful synthesis of the nice integrative topics presented here.

I would say that the introduction could use more specifics about the stable isotopes at vents where there can be pretty significant variation in both C and N at the base of the food web due to symbionts (often negative N) in contrast to other inputs, plus the relatively high N of phytodetritus. It adds a dimension to the isotopic analyses in other systems and without this mention may confuse the reader until it is discussed in the discussion. Simply a sentence or two in the introduction could help the readers have a better foundation for this. L511 in the discussion does point towards it but without specific examples. Simply adding a sentence at line 119 saying that these different sources of primary production vary in del 15 N making clear trophic analysis more complex would be one way to do that.

Results – I am torn on the use of the 3-dimensional space for extrapolating up the total density of fauna. I believe the numbers could be important but really it is a two-2 surface area that is expanded up by ecosystem engineers but limited by the energy input and space, which is more 2-d. At a minimum, a statement and comparison of the 2-d abundance would be an important comparison, especially when comparing different habitats where the increased area of tube worms will decrease the extrapolation up (i.e. fewer fauna per m2 but with a lower height measurement will come up with a much higher ind. Mˆ-3 value than one that had a higher density on the per m2 but since you measured a larger area will be fewer on the m3 metric).

I found the treatment of the isotope data not comprehensive enough to support the conclusions made. Specifically, L 362 identifies a shift from bactivorous to predator guild, but which species belong to which? How is this shown by these data? I also question whether the term "trophic network" is appropriate. Really these data are just presented and then scaled by biomass, which I like, but is not a trophic network per se. Either modify the term or expand the analyses performed to look more at connections. I am not sure that a more in depth analysis would be possible with the heterogeneous and every shifting baseline caused by the diversity of microbial communities so that is not what I am recommending, but instead I would avoid the term trophic network.

Figures 1-3 are very nice. Figures 4 and 5 have too small of font on the axis and the grey background clutters the visuals, especially when numbered, also too small. I do not consider these two figures ready for publication. The grey should be removed, the lines within the text should be removed and ideally a key with the colors and the species should be included so the reader is not forced to delve heavily into the figure legend to know what they are looking at.

Small suggestions:

L37 "Fairly" long tubes comes across as vague. Since the actual values are known, please just include them. L122 – I would suggest adding in "average rate of +3.2" as that is a mean of multiple, often highly variable values. L144 – The sentence that starts on this line seems out of place in this paragraph. It should be either removed or rephrased as to why this builds upon what was said before. L294-296 It seems that these should either be reported in dmˆ3 or with a different number of significant figures as there was not a mˆ3 counted. I understand why mˆ3 was used so suggest just 17 x 10ˆ6 etc. but also comparing them in a m-2 context. L290- again question whether the right number of significant digits is used on the percentage Line 361 "contributed – 16.4%..." L386 – also sampling approaches. Any of the sampling that has occurred with a mussel pot or a Bushmaster could also lead to differences in diversity simply due to methodology. In addition, not suctioning the area could also lead to lesser diversity

in other studies. L414 – I question whether a trophic network is the right word here considering the analyses done.

---

## Author Comment (AC1) · 23 Feb 2018

We are grateful to Dr. Boschen for her constructive suggestions that helped us improve our manuscript significantly.

Please find attached our response to Rachel Boschen's comments.

Please also note the supplement to this comment:
https://www.biogeosciences-discuss.net/bg-2017-411/bg-2017-411-AC1-supplement.pdf

[Figure]

**Table 1.** Trophic guild and nutritional modes of macrofaunal species associated with the *Ridgeia piscesae* tubeworm assemblages of the Grotto hydrothermal edifice (Main Endeavour Field, Juan de Fuca Ridge). An asterisk (*) marks the original description of the species.

| Species | Trophic guild - Nutritional mode | Reference(s) |
|---|---|---|
| **Annelida** | | |
| **Polychaeta** | | |
| Siboglinidae | | |
| *Ridgeia piscesae* | Symbiotic | Jones, 1985*; Southward et al., 1995; Bergquist et al., 2007; This study |
| Maldanidae | | |
| *Nicomache venticola* | Bacterivore - Surface deposit feeder or grazer | Blake and Hilbig, 1990*; Bergquist et al., 2007; This study |
| Dorvilleidae | | |
| *Ophryotrocha globopalpata* | Predator | Blake and Hilbig, 1990*; Bergquist et al., 2007; This study |
| Orbiniidae | | |
| *Berkeleyia* sp. nov. | Scavenger/detritivore - Suspension feeder | Jumars et al., 2015; This study |
| Hesionidae | | |
| *Hesiospina* sp. nov.[1] | Predator | Bonifácio et al., *in preparation*; This study |
| Phyllodocidae | | |
| *Protomystides verenae* | Predator | Blake and Hilbig, 1990*; Bergquist et al., 2007; This study |
| Polynoidae | | |
| *Branchinotogluma tunnicliffeae* | Predator | Pettibone, 1988*; Bergquist et al., 2007; This study |
| *Branchinotogluma* sp. | Predator | - |
| *Lepidonotopodium piscesae* | Predator | Pettibone, 1988*; Levesque et al., 2006; Bergquist et al., 2007; This study |
| *Levensteiniella kincaidi* | Predator | Pettibone, 1985*; Bergquist et al., 2007; This study |
| Sigalionidae | | |
| *Pholoe courtneyae* | Predator | Blake, 1995*; Sweetman et al., 2013 |
| Syllidae | | |
| *Sphaerosyllis ridgensis* | Predator | Blake and Hilbig, 1990*; Bergquist et al., 2007; This study |
| Aivinellidae | | |
| *Paralvinella dela* | Bacterivore - Surface deposit feeder or grazer; suspension feeder | Detinova et al., 1988*; This study |
| *Paralvinella palmiformis* | Bacterivore - Surface deposit feeder or grazer; suspension feeder | Desbruyères and Laubier, 1986*; Desbruyères and Laubier, 1991; Levesque et al., 2003; This study |
| *Paralvinella pandorae* | Bacterivore - Surface deposit feeder or grazer; suspension feeder | Desbruyères and Laubier, 1986*; Desbruyères and Laubier, 1991; Levesque et al., 2003; This study |
| *Paralvinella sulfincola* | Bacterivore - Surface deposit feeder or grazer; suspension feeder | Tunnicliffe et al., 1993*; Levesque et al., 2003; This study |
| Ampharetidae | | |
| *Amphisamytha carldarei* | Scavenger/detritivore - Surface deposit feeder or grazer | Stiller et al., 2013*; McHugh and Tunnicliffe, 1994; Bergquist et al., 2007; This study |
| Cteondilidae | | |
| *Raricirrus* sp. | Bacterivore - Surface deposit feeder or grazer | Jumars et al., 2015 |
| Spionidae | | |
| *Prionospio* sp. | Bacterivore - Surface deposit feeder or grazer | Jumars et al., 2015 |
| **Mollusca** | | |
| **Aplacophora** | | |
| Simrothiellidae | | |
| *Helicoradomenia juani* | Predator | Scheltema and Kuzirian, 1991*; Bergquist et al., 2007; This study |
| **Gastropoda** | | |
| Buccinidae | | |
| *Buccinum thermophilum* | Scavenger/detritivore - Surface deposit feeder or grazer | Harasewych and Kantor, 2002*; Martell et al., 2002; This study |
| Provannidae | | |
| *Provanna variabilis* | Bacterivore - Surface deposit feeder or grazer | Warén and Bouchet, 1986*; Bergquist et al., 2007; This study |
| Peltospiridae | | |
| *Depressigyra globulus* | Bacterivore - Surface deposit feeder or grazer | Warén and Bouchet, 1989*; Bergquist et al., 2007; This study |
| Clypeosectidae | | |
| *Clypeosectus curvus* | Predator | McLean, 1989*; Bergquist et al., 2007; This study |
| Lepetodrilidae | | |
| *Lepetodrilus fucensis* | Symbiotic; Bacterivore - Surface deposit feeder or grazer; suspension feeder | McLean, 1988*; Fox et al., 2002; Bates et al., 2007; Bergquist et al., 2007; This study |
| **Arthropoda** | | |
| **Arachnida** | | |
| Halacaridae | | |
| *Copidognathus papillatus* | Predator | Krantz, 1982*; Bergquist et al., 2007; This study |
| **Amphipoda** | | |
| Alicellidae | | |
| *Paralicella cf. vaporalis* | - | Barnard and Ingram, 1990* |
| Calliopiidae | | |
| *Oradarea cf. walkeri* | - | Shoemaker, 1930* |
| *Leptamphopus* sp. | - | - |
| **Crustacea** | | |
| Philomedidae | | |
| *Euphilomedes climax* | Bacterivore - Surface deposit feeder or grazer | Kornicker, 1991*; This study |
| Cytheruridae | | |
| *Xylocythere* sp. nov.[2] | Bacterivore - Surface deposit feeder or grazer | Tanaka et al., *in preparation*; Maddocks and Steineck, 1987; This study |
| **Pycnogonida** | | |
| Ammotheidae | | |
| *Sericosura verenae* | Bacterivore - Surface deposit feeder or grazer | Child, 1987*; Bergquist et al., 2007; This study |
| *Sericosura venticola* | Bacterivore - Surface deposit feeder or grazer | Child, 1987*; Bergquist et al., 2007; This study |
| *Sericosura cf. dissita* | Bacterivore - Surface deposit feeder or grazer | Child, 2000*; This study |
| **Nemertea** | | |
| **Unidentified** | Predator | - |
| **Echinodermata** | | |
| Ophiuridae | - | - |

[1]This species refers to *Hesiospina legendrei* that is currently under description (Bonifácio et al., *in preparation*)
[2]This species refers to *Xylocythere samsamae* that is currently under description (Tanaka et al., *in preparation*)

**Fig. 1.** Table 1 - Trophic guild and nutritional mode

**Table 2.** Percentage abundance x 100 (% Ab.), faunal density (D. ind m⁻²), volume (V. ind m⁻³) and relative biomass x 100 (% Biom.) of the different macrofaunal taxa (>250 µm) identified in the 6 sampling units (S1 to S6) on the Grotto edifice. The taxa were identified to the lowest possible taxonomical level.

| Species | S1 - Assemblage V low-flow | | | | S2 - Assemblage V low-flow | | | | S3 - Assemblage V low-flow | | | | S4 - Assemblage IV | | | | S5 - Assemblage V low-flow | | | | S6 - Assemblage III | | | |
|---|---|---|---|---|---|---|---|---|---|---|---|---|---|---|---|---|---|---|---|---|---|---|---|---|
| | % Ab. | D. (ind m⁻²) | V. (ind m⁻³) | % Biom. | % Ab. | D. (ind m⁻²) | V. (ind m⁻³) | % Biom. | % Ab. | D. (ind m⁻²) | V. (ind m⁻³) | % Biom. | % Ab. | D. (ind m⁻²) | V. (ind m⁻³) | % Biom. | % Ab. | D. (ind m⁻²) | V. (ind m⁻³) | % Biom. | % Ab. | D. (ind m⁻²) | V. (ind m⁻³) | % Biom. |
| **Annelida** | | | | | | | | | | | | | | | | | | | | | | | | |
| **Polychaeta** | | | | | | | | | | | | | | | | | | | | | | | | |
| Maldanidae | | | | | | | | | | | | | | | | | | | | | | | | |
| Nicomache venticola | 0.1 | 8 | 1604.5 | 0.2 | 0.1 | 14.1 | 7033.6 | 0.1 | 0.2 | 35.5 | 6363.3 | 1.1 | 0 | 0 | 0 | 0 | 0 | 0 | 0 | 0 | 0 | 0 | 0 | 0 |
| Dorvilleidae | | | | | | | | | | | | | | | | | | | | | | | | |
| Ophryotrocha globopalpata | 0.1 | 4.2 | 849.5 | <0.1 | 0.1 | 9.8 | 4893 | <0.1 | 1.3 | 202 | 36241.1 | <0.1 | <0.1 | 1.8 | 934.6 | <0.1 | <0.1 | 10 | 1851.9 | <0.1 | 0 | 0 | 0 | 0 |
| Orbiniidae | | | | | | | | | | | | | | | | | | | | | | | | |
| Berkeleyia sp. nov. | 0 | 0 | 0 | 0 | 0 | 0 | 0 | 0 | <0.1 | 6.3 | 1137.8 | <0.1 | 0 | 0 | 0 | 0 | 0 | 0 | 0 | 0 | 0 | 0 | 0 | 0 |
| Hesionidae | | | | | | | | | | | | | | | | | | | | | | | | |
| Hesiospina sp. nov. | 0.1 | 9.8 | 1982.1 | <0.1 | 0.1 | 10.4 | 5198.8 | <0.1 | 0.1 | 14.3 | 2570.6 | <0.1 | 0 | 0 | 0 | 0 | 0 | 0 | 0 | 0 | 0 | 0 | 0 | 0 |
| Phyllodocidae | | | | | | | | | | | | | | | | | | | | | | | | |
| Protomystides verenae | <0.1 | 0.5 | 94.4 | <0.1 | <0.1 | 0.6 | 305.8 | <0.1 | <0.1 | 0.2 | 42.1 | <0.1 | 0 | 0 | 0 | 0 | 0 | 0 | 0 | 0 | 0 | 0 | 0 | 0 |
| Polynoidae | | | | | | | | | | | | | | | | | | | | | | | | |
| Branchinotogluma tunnicliffeae | <0.1 | 0.7 | 141.6 | <0.1 | 0.1 | 9.2 | 4587.2 | <0.1 | <0.1 | 2.6 | 463.6 | <0.1 | 0.2 | 47.4 | 25233.7 | 0.2 | <0.1 | 20 | 3703.7 | <0.1 | 0.3 | 36.4 | 490 | <0.1 |
| Branchinotogluma sp. | 0 | 0 | 0 | 0 | 0 | 0 | 0 | 0 | 0 | 0 | 0 | 0 | <0.1 | 1.8 | 934.6 | <0.1 | 0 | 0 | 0 | 0 | 0 | 0 | 0 | 0 |
| Lepidonotopodium piscesae | 0 | 0 | 0 | 0 | <0.1 | 1.2 | 611.6 | <0.1 | <0.1 | 3.8 | 674.3 | <0.1 | 0.2 | 43.9 | 21064.5 | 0.9 | 0.1 | 50.1 | 9259.3 | 0.2 | 0.2 | 10.9 | 266.7 | 0.1 |
| Levensteiniella kincaidi | 0.1 | 8.2 | 1651.7 | 0.1 | 0.1 | 10.4 | 5198.8 | <0.1 | <0.1 | 4 | 716.4 | <0.1 | <0.1 | 3.5 | 1869.2 | <0.1 | <0.1 | 20 | 3703.7 | <0.1 | 0 | 0 | 0 | 0 |
| Sigalionidae | | | | | | | | | | | | | | | | | | | | | | | | |
| Pholoe courtneyae | <0.1 | 0.2 | 47.2 | <0.1 | 0 | 0 | 0 | 0 | 0 | 0 | 0 | 0 | 0 | 0 | 0 | 0 | 0 | 0 | 0 | 0 | 0 | 0 | 0 | 0 |
| Syllidae | | | | | | | | | | | | | | | | | | | | | | | | |
| Sphaerosyllis ridgensis | 3.6 | 257.6 | 51911.3 | <0.1 | 1 | 170.2 | 85015.1 | <0.1 | 1.7 | 262.6 | 47113.4 | <0.1 | 0 | 0 | 0 | 0 | 0.1 | 90.1 | 16666.7 | <0.1 | 0.1 | 5.5 | 133.3 | <0.1 |
| Alvinellidae | | | | | | | | | | | | | | | | | | | | | | | | |
| Paralvinella dela | 0 | 0 | 0 | 0 | 0 | 0 | 0 | 0 | <0.1 | 0.7 | 126.4 | <0.1 | 3.3 | 726.4 | 386915.9 | 6.8 | 0.2 | 110.2 | 20370.4 | 0.8 | 0.7 | 43.6 | 1066.7 | 1.4 |
| Paralvinella palmiformis | 0 | 0 | 0 | 0 | 0.1 | 10.4 | 5198.8 | <0.1 | 0.1 | 10.3 | 1854.2 | 0.1 | 0.1 | 17.6 | 9345.8 | <0.1 | 0.1 | 70.1 | 12963 | <0.1 | 5.4 | 354.3 | 8666.7 | <0.1 |
| Paralvinella pandorae | <0.1 | 0.7 | 141.6 | <0.1 | <0.1 | 2.5 | 1223.2 | <0.1 | 0.1 | 17.4 | 3118.4 | <0.1 | 0.6 | 122.8 | 65420.6 | 0.5 | <0.1 | 10 | 1851.9 | <0.1 | 0.2 | 10.9 | 266.7 | 0.1 |
| Paralvinella sulfincola | 0 | 0 | 0 | 0 | 0 | 0 | 0 | 0 | <0.1 | 6.6 | 1179.9 | 0.1 | 0 | 0 | 0 | 0 | 0 | 0 | 0 | 0 | 0 | 0 | 0 | 0 |
| Amphisamytha | | | | | | | | | | | | | | | | | | | | | | | | |
| Amphisamytha carldarei | 26.8 | 1932.1 | 389334.6 | 0.5 | 24.5 | 3167.5 | 1681651.4 | 0.5 | 34.8 | 5378.8 | 964938.9 | 1.8 | 0.4 | 80.7 | 42990.7 | <0.1 | 5.1 | 3004.6 | 555555.6 | 0.2 | 7.4 | 479.6 | 11733.3 | 0.1 |
| Ctenodrilidae | | | | | | | | | | | | | | | | | | | | | | | | |
| Raricirrus sp. | <0.1 | 0.2 | 47.2 | <0.1 | 0 | 0 | 0 | 0 | <0.1 | 1.9 | 337.1 | <0.1 | 0 | 0 | 0 | 0 | 0 | 0 | 0 | 0 | 0 | 0 | 0 | 0 |
| Spionidae | | | | | | | | | | | | | | | | | | | | | | | | |
| Prionospio sp. | <0.1 | 2.1 | 424.7 | <0.1 | 0 | 0 | 0 | 0 | <0.1 | 0.9 | 168.6 | <0.1 | 0 | 0 | 0 | 0 | 0 | 0 | 0 | 0 | 0 | 0 | 0 | 0 |
| **Mollusca** | | | | | | | | | | | | | | | | | | | | | | | | |
| **Aplacophora** | | | | | | | | | | | | | | | | | | | | | | | | |
| Simrothiellidae | | | | | | | | | | | | | | | | | | | | | | | | |
| Helicoradomenia juani | 3.7 | 263.7 | 53138.3 | <0.1 | 3.4 | 461.1 | 230275.2 | <0.1 | 5.6 | 861.9 | 154614.4 | 0.14 | 0.1 | 17.6 | 9345.8 | <0.1 | 1.3 | 791.2 | 146296.3 | <0.1 | 2 | 130.8 | 3200 | <0.1 |
| **Gastropoda** | | | | | | | | | | | | | | | | | | | | | | | | |
| Buccinidae | | | | | | | | | | | | | | | | | | | | | | | | |
| Buccinum thermophilum | 0.1 | 9.8 | 1982.1 | 3.2 | 0.1 | 18.4 | 9174.3 | 1.6 | <0.1 | 2.4 | 421.4 | 0.4 | 0 | 0 | 0 | 0 | 0.1 | 50.1 | 9259.3 | 5.8 | 0 | 0 | 0 | 0 |
| Provannidae | | | | | | | | | | | | | | | | | | | | | | | | |
| Provanna variabilis | 1.9 | 137 | 27607.4 | 2.2 | 1.6 | 224.8 | 112232.4 | 1.3 | 4.5 | 694.8 | 124652.3 | 6.8 | 0.3 | 61.4 | 32710.3 | 0.3 | 8.6 | 5117.8 | 946296.3 | 10 | 1.8 | 114.5 | 2800 | 0.6 |
| Peltospiridae | | | | | | | | | | | | | | | | | | | | | | | | |
| Depressigyra globulus | 10.8 | 779.9 | 157149.6 | 1.1 | 14.5 | 1986 | 991743.1 | 1.1 | 20.1 | 3105.4 | 557100.7 | 2.9 | 51.4 | 11294.7 | 6015887.9 | 13 | 44 | 26179.8 | 4840740.7 | 21 | 23.2 | 1515.1 | 37066.7 | 1.5 |
| Clypeosectidae | | | | | | | | | | | | | | | | | | | | | | | | |
| Clypeosectus curvus | 0.5 | 34.7 | 6984.4 | 0.1 | 0.2 | 20.2 | 10091.7 | <0.1 | <0.1 | 6.8 | 1222.1 | <0.1 | 0 | 0 | 0 | 0 | 0.1 | 50.1 | 9259.3 | <0.1 | 0 | 0 | 0 | 0 |
| Lepetodrilidae | | | | | | | | | | | | | | | | | | | | | | | | |
| Lepetodrilus fucensis | 42.7 | 3083.1 | 621283.6 | 12.5 | 39.5 | 5429.4 | 2711315 | 10.1 | 22.8 | 3519.5 | 631394.9 | 11.5 | 43 | 9457.6 | 5037383.2 | 19.4 | 30.1 | 17907.2 | 3311111.1 | 18.1 | 55 | 3586.1 | 87733.3 | 18.2 |
| **Arthropoda** | | | | | | | | | | | | | | | | | | | | | | | | |
| **Arachnida** | | | | | | | | | | | | | | | | | | | | | | | | |
| Halacaridae | | | | | | | | | | | | | | | | | | | | | | | | |
| Copidognathus papillatus | 6.9 | 349.6 | 70457.8 | <0.1 | 1.5 | 211.9 | 105810.4 | <0.1 | 4.9 | 754.7 | 135398.2 | <0.1 | <0.1 | 8.8 | 4672.9 | <0.1 | <0.1 | 60.1 | 11111.1 | <0.1 | 0 | 0 | 0 | 0 |
| **Amphipoda** | | | | | | | | | | | | | | | | | | | | | | | | |
| Alicellidae | | | | | | | | | | | | | | | | | | | | | | | | |
| Paralicella cf. vaporalis | <0.1 | 1.6 | 330.3 | <0.1 | <0.1 | 6.1 | 3058.1 | <0.1 | <0.1 | 3.1 | 547.8 | <0.1 | 0 | 0 | 0 | 0 | 0 | 0 | 0 | 0 | 0 | 0 | 0 | 0 |
| Calliopiidae | | | | | | | | | | | | | | | | | | | | | | | | |
| Oradarea cf. walkeri | <0.1 | 4.7 | 943.8 | <0.1 | 0 | 0 | 0 | 0 | 0 | 0 | 0 | 0 | 0 | 0 | 0 | 0 | 0 | 0 | 0 | 0 | 0 | 0 | 0 | 0 |
| Leiopompharpus sp. | 0 | 0 | 0 | 0 | 0 | 0 | 0 | 0 | 0 | 0 | 0 | 0 | <0.1 | 3.5 | 1869.2 | <0.1 | 0 | 0 | 0 | 0 | 0 | 0 | 0 | 0 |
| **Crustacea** | | | | | | | | | | | | | | | | | | | | | | | | |
| Philomedidae | | | | | | | | | | | | | | | | | | | | | | | | |
| Euphilomedes climax | 0.7 | 52 | 10476.6 | <0.1 | 10.9 | 1502.2 | 750152.9 | 0.1 | 0.2 | 23.3 | 4171.9 | <0.1 | 0.3 | 70.2 | 37383.2 | <0.1 | 9.1 | 5378.2 | 994444.4 | <0.1 | 3.3 | 218 | 5333.3 | <0.1 |
| Cytheruridae | | | | | | | | | | | | | | | | | | | | | | | | |
| Xylocythere sp. nov. | 2.9 | 206.8 | 41670.6 | <0.1 | 1.4 | 188 | 93883.8 | <0.1 | 2.7 | 413 | 74083.4 | <0.1 | 7 | 413 | 3738.3 | <0.1 | 0.3 | 170.3 | 31481.5 | <0.1 | 0.1 | 5.5 | 133.3 | <0.1 |
| **Pycnogonida** | | | | | | | | | | | | | | | | | | | | | | | | |
| Ammotheidae | | | | | | | | | | | | | | | | | | | | | | | | |
| Sericosura verenae | 0.6 | 42.6 | 8589.0 | <0.1 | 0.6 | 84.5 | 42201.8 | 0.5 | 0.7 | 113.5 | 20354 | 0.8 | 0.1 | 24.6 | 13084.1 | 0.2 | 0.6 | 360.6 | 66666.7 | 1.2 | 0.4 | 27.3 | 666.7 | 0.2 |
| Sericosura venticola | 0.2 | 15.2 | 3067.5 | <0.1 | 0.1 | 8 | 3975.5 | 0.5 | <0.1 | 1.9 | 337.1 | 0.8 | <0.1 | 1.8 | 934.6 | <0.1 | 0 | 0 | 0 | 0 | 0 | 0 | 0 | 0 |
| Sericosura cf. dissita | 0.1 | 5.6 | 1132.6 | <0.1 | 0.1 | 1.2 | 611.6 | 0.5 | <0.1 | 0.7 | 126.4 | 0.8 | 0 | 0 | 0 | 0 | 0 | 0 | 0 | 0 | 0 | 0 | 0 | 0 |
| **Nemertea** | | | | | | | | | | | | | | | | | | | | | | | | |
| **Unidentified** | <0.1 | 1.2 | 236.0 | <0.1 | 0 | 0 | 0 | 0 | <0.1 | 2.8 | 505.7 | <0.1 | 0 | 0 | 0 | 0 | 0 | 0 | 0 | 0 | 0 | 0 | 0 | 0 |
| **Echinodermata** | | | | | | | | | | | | | | | | | | | | | | | | |
| **Ophiuridae** | <0.1 | 0.2 | 47.2 | <0.1 | 0 | 0 | 0 | 0 | <0.1 | 0.5 | 84.3 | <0.1 | 0 | 0 | 0 | 0 | 0 | 0 | 0 | 0 | 0 | 0 | 0 | 0 |

**Fig. 2.** Table 2 - Species List

**Table 3.** Univariate measures of macrofaunal community structure associated with *Ridgeia piscesae* tubeworm bushes on the Grotto edifice: Sample Surface Area (SSA), Tube Height (TH), Tube Diameter (TD), volume (V), species richness (S), exponential of Shannon entropy (D), Simpson's diversity index (1-$\lambda'$) and Pielou's evenness (J').

| Sample | SSA (m$^2$) | TH (cm) ± sd | TD (cm) ± sd | TSA (m$^2$) | V (m$^3$) | S | D | 1-$\lambda'$ | J' |
|---|---|---|---|---|---|---|---|---|---|
| S1 | 0.12 | 17.14 ± 6.83 | 4.72 ± 0.82 | 4.27 | 0.02 | 28 | 5.377 | 0.728 | 0.505 |
| S2 | 0.06 | 5.32 ± 2.47 | 2.24 ± 0.42 | 1.63 | 3.4×10$^{-3}$ | 24 | 5.398 | 0.749 | 0.531 |
| S3 | 0.12 | 19.91 ± 9.31 | 5.48 ± 1.13 | 4.26 | 0.02 | 31 | 6.053 | 0.778 | 0.524 |
| S4 | 0.02 | 7.15 ± 2.45 | 2.48 ± 0.57 | 0.57 | 1.1×10$^{-3}$ | 19 | 2.605 | 0.55 | 0.325 |
| S5 | 0.02 | 3.46 ± 0.85 | 1.91 ± 0.34 | 0.10 | 5.4×10$^{-4}$ | 19 | 4.348 | 0.697 | 0.499 |
| S6 | 0.01 | 6.15 ± 2.80 | 2.14 ± 0.43 | 0.18 | 7.5×10$^{-4}$ | 14 | 3.998 | 0.633 | 0.525 |

**Fig. 3.** Table 3- Sample descriptors

**Supplement:**

**Responses to Editor and Referees**

**Rachel Boschen**

**"Biodiversity and trophic ecology of hydrothermal vent fauna associated with tubeworm assemblages on the Juan de Fuca Ridge"**

**bg-2017-411**

**I. General comments**

*"Overall the manuscript is well written and the concepts discussed are interesting. The approach is not novel; papers have been written on the community structure and stable isotope characters of these species previously. However, the extension of this work to the concept of a 'trophic network' through linkages to biomass has not been done in this location previously and the perspectives it provides are interesting. Some areas of the manuscript require clarification and there are questions about some aspects of the approach. I couldn't find the tables referenced in the manuscript: some of my concerns may be addressed by being able to evaluate these."*

Author and co-authors: **We are grateful to Dr. Boschen for her constructive suggestions that helped us improve our manuscript significantly. We addressed all comments and issues listed below.**

**II. Specific comments**

*"Species list and abundances don't seem to be provided, although they are mentioned as being in Table 2. These should be made available, at least in supplementary format. Species list provision is particularly important for this study as all the samples were collected within the Endeavour Hydrothermal Vents Marine Protected Area (EHV MPA); species occurrence data would be very useful to the managing authority Department of Fisheries and Oceans (DFO). No mention of EHV MPA is made within the manuscript."*

Author and co-authors: **The tables have been prepared and submitted but it seems that a problem might have occurred during the submission/revision process. The other reviewer did not seem to have the same problem. A sentence specifying the MPA status of Endeavour Hydrothermal Vents has been added on lines 178-182 pages 6-7:** *"This vent field is the world's first hydrothermal marine protected area (Devey et al., 2007; Tunnicliffe and*

*Thomson, 1999) and was selected as a target site for the deep-sea cabled observatory Ocean Networks Canada, providing unprecedented opportunities to better understand vent ecology."*

*"36 macrofaunal taxa were identified to the species level" (line 33); "35 macrofaunal taxonomic groups" (line 287 & 367): which is it - 36 species, or 35 taxa? Please clarify."*

**Author and co-authors: A total of 36 macrofaunal species, including *Ridgeia piscesae*, were identified to the species level. On lines 293-294 page 10, we modified the sentence to clarify that the engineering species *R. piscesae* is not included in the species count:** *"[…] 35 macrofaunal species (excluding Ridgeia piscesae) were identified in the six tubeworm bushes […]".* **Consequently, 35 macrofaunal species and the engineering species *R. piscesae* add up to 36 macrofaunal species. In addition, we modified the term "taxa" to "species" because all organisms were sorted to the species level.**

*"Line 37: "fairly long tubes". Please define, how long?"*

**Author and co-authors: We modified the sentence line 38 page 2 and added the real measurement value:** *"[…] tubeworm aggregations characterized by the longest tubes (18.5 ± 3.3 cm).".*

*"Lines 98 – 101: limited behavioural observations and gut contents analyses, compared to wide use of stable isotopes. For JdFR vents, there are many observations on behaviour, especially feeding, and some on gut contents, in the original species descriptions. True, there are few studies specifically addressing feeding behaviour or gut contents, compared to dedicated stable isotope studies. However, stable isotopes are just one of many tools to evaluate feeding, and like other techniques, stable isotopes have their limitations. It would be helpful to have a few sentences acknowledging the pros & cons of stable isotopes compared to other techniques available to assess feeding and trophic structure."*

**Author and co-authors: As suggested by the Reviewer, we added few sentences about this point, lines 99-108 page 4:** *"Stable isotopes analysis is an important and efficient tool in studying trophic ecology and offers many advantages over traditional methods (behavioural observations, stomach content analyses), providing time-integrated overview of animal's diet over a long timescale. Nevertheless, the physiology of marine invertebrates is poorly*

*documented, resulting to a lack of precision on the turnover rate of organism's tissues and therefore, the rate at which a consumer integrates the isotopic signal, leading to an uncertainty about trophic-step fractionation (isotopic enrichment between preys and predators). Moreover, although trophic inferences using stable isotopes require the characterization of basal sources, this remains difficult in the hydrothermal environment due to sampling technological constraint."*.

*"Line 115: "despite the relatively low diversity of the deep-sea community" - do you mean low diversity of the hydrothermal community? Lines 63-64: "[hydrothermal vent communities have] low species diversity compared with adjacent deep-sea and coastal benthic communities". Please clarify."*

**Author and co-authors: The original sentence was incorrect, "deep-sea" was modified to "vent", line 123 page 5:** *"Despite the relatively low diversity of vent communities, […]".*

*"Line 122: Urcuyo et al. 2003. This is not the first reference to Ridgeia piscesae populations along the JdFR; please consider replacing or supplementing with an earlier reference. For example, the Revision of Ridgeia was published by Southward et al. in 1995; records of Ridgeia along the JdFR will extend to before this reference. "Southward et al. (1995) Revision of the species of Ridgeia from northeast Pacific hydrothermal vents, with a redescription of Ridgeia piscesae Jones (Pogonophora: Obturata = Vestimentifera. Canadian Journal of Zoology. 73: 282-295."*

**Author and co-authors: We modified the reference "Urcuyo et al. 2003" to "Southward et al. 1995", lines 137-138 page 5:** *"siboglinid polychaete Ridgeia piscesae forming dense faunal assemblages in areas of high to low fluid flux activity (Southward et al., 1995)."*

*"Line 153: Lelievre et al. 2017. There is an earlier study on tidal influence on R. piscesae that was not referenced in Lelievre et al. 2017 and should probably be referenced here: Johnson & Tunnicliffe (1985) Time-series measurements of hydrothermal activity on northern Juan de Fuca Ridge. Geophysical Research Letters 12: 10, 685-688."*

**Author and co-authors: The paper of Johnson & Tunnicliffe was indeed the first to hypothesize a possible influence of tides on R. piscesae. Here, the paragraph is written in the context of a seafloor observatory, and Lelièvre et al. 2017 was cited to highlight the**

**role of the observatory in acquiring new types of data in the vent environment. Indeed, Cuvelier et al. 2014 and Lelièvre et al. 2017 were the first papers to directly measure the influence of tides and surface storms on vent species behaviour/abundance using the observatory high resolution imagery data. This sentence was included to show the ecological importance and valuable information provided by observatory data.**

*"Line 177: "intense". Intense in terms of what? This could probably be removed so the sentence reads "The MEF… is the most active of the five hydrothermal fields…""*

**Author and co-authors: The sentence has been modified as suggested, line 176 page 6:** *"The MEF (Fig. 1c) is the most active of the five hydrothermal fields, […]"*.

*"Line 183: "Like many other MEF hydrothermal edifices…" This only has one reference, Sarrazin et al. 1997. This reference only characterises one hydrothermal structure, S & M. You need more references here to support the 'many' in your sentence; Govenar et al. 2002, Bergquist et al. 2007 and Urcuyo et al. 2007 are all possible additions. The latter is not in your reference list: Urcuyo et al. (2007) Growth and longevity of the tubeworm Ridgeia piscesae in the variable diffuse flow habitats of the Juan de Fuca Ridge. Marine Ecology Progress Series 344: 143-157."*

**Author and co-authors: As suggested by the reviewer, references have been added lines 185-188 page 7:** *"Like many other MEF hydrothermal edifices, the site is largely colonized by dense assemblages of Ridgeia piscesae (Polychaeta, Siboglinidae) with their associated fauna (Bergquist et al., 2007; Govenar et al., 2002; Sarrazin et al., 1997; Urcuyo et al., 2007)."*

*"Lines 203-206: macrofauna and meiofauna sampling. If macrofaunal specimens are ">250 um" and meiofauna are "<63 um" then what are the fauna that are <250 um but > 63 um? The definition of macrofauna is variable, but 250 um is on the smaller end. 0.5 mm or 1 mm sieves are more common for macrofauna retention. Meiofauna would usually be considered to be 63 um – 0.5 mm: <63 um would be microfauna."*

**Author and co-authors: We modified the sentence and corrected this oversight, lines 208-209 page 7:** *"Macrofaunal specimens (>250 µm) were preserved in 96 % ethanol and meiofauna (250 µm > x > 63 µm) in 10 % seawater formalin."*.

*"Lines 209-212: trophic guilds. If trophic guilds were assigned to species prior to your stable isotope analysis, there needs to be a table detailing which species were assigned to different guilds, preferably with the reference(s) used for the designation and an indication if species level of higher was available. From the way this is written in the methods, it almost sounds like a circular argument: literature was used to assign trophic guilds (which will mostly contain stable isotope information) and then stable isotopes were used to assess the validity of these groups. There needs to be more detail here as to how the 'trophic guild' information was obtained and how it was applied."*

**Author and co-authors:**  **We compiled data from the literature in order to compare our isotopic data. We clarify this point lines 213-217 page 8:** *"Trophic guilds from the literature (symbiont host, bacterivore, scavenger/detritivore or predator) were compiled for each vent species (Table 1). For species with unknown diets, the compilation was based on trophic guilds identified from closely related species (within the same family). These were then compared to our isotopic results (see below)."* **As suggested by the reviewer, we provided also a table with the trophic guild and nutritional mode of each species, as well as their respective references (Table 1).**

*"Line 215-222: density of individuals per m$^{-2}$. This is presumably based on the surface area of the sulphide chimney the tubeworms were attached to. But it is not really the surface area for the fauna – that would be the tubeworm surface area. I would be very cautious about using ind. m$^{-2}$ based on the chimney area sampled. Equally, as the basis for ind. m$^{-3}$ I would be cautious. I appreciate this volume takes into consideration average tubeworm length, but it doesn't take any account the width of the tubes or how twisted and intermeshed they are, which all contribute to the habitat area the tubeworms provide. Tsurumi and Tunnicliffe (2001), Tsurumi and Tunnicliffe (2003) & Marcus and Tunnicliffe (2009) all use tubeworm area standardised abundances for this reason."*

**Author and co-authors:** **We agree that density at the base does not reflect the reality of the assemblage, which is why we calculated a volume. However, we appreciate that the volume indicated is somewhat biased because of the space occupied by the siboglinids themselves. We thus now modified the m² at the base by the total tubes' surface area calculated under the assumption that the tubes are modelled by a cylinder shape. For this, we measured the length of each tube by taking into account the diameter. We obtained**

**the developed surface (surface of tubeworms available for the fauna) in m$^2$. The methodology is now explained lines 220-227 page 8:** *"The Ridgeia piscesae tubes create a three-dimensional (3D) structure for other vent animals to colonize. An estimation of the tubeworm surface area available for the fauna and the volume for each assemblage provided a proxy for habitat complexity. Assuming that the tubes are cylinders erected vertically, tubeworm surface area was estimated by measuring lengthwise and crosswise (i.e. diameter) 10 % of the tubeworm tubes randomly selected. For each tubeworm assemblage, species density is therefore expressed in number of individuals per square meter of tubeworm surface (ind m$^{-2}$) to account for this 3D space (Marcus et al., 2009; Tsurumi and Tunnicliffe, 2001, 2003)."* **and species densities (in ind m$^{-2}$) were added in Table 3.**

*"Lines 223-226: biomass estimates. I find these lines a little unclear... for the total dry mass of each species, did you dry all individuals of those species? Did you just dry the 3 – 10 individuals that you had randomly selected? If just the selected individuals, were these dried separately or together? What do you mean by "absolute biomass of each species was calculated by multiplying the relative biomass by the abundance of each species"? Did you take the mean weight of your randomly selected individuals and multiply this by your abundance? Please provide some clarification on the biomass estimates section."*

**Author and co-authors: We clarified this point in the manuscript lines 229-232, page 8. It now reads:** *"Biomass estimates were obtained from the total dry mass (DM) of a random sample of a few individuals (3-10) for each species. DM of each species corresponds to the mean of individual masses obtained after drying each individual at 80°C for 48 h, multiplied by the abundance of each species."*.

*"Lines 230-231: gut content. I'm not clear what you mean here by "in the case of intermediate-sized specimens, the gut content was removed". Does this mean you did the stable isotope analysis on the whole individual, minus its gut? Or that you did the stable isotope work on the gut contents? Please clarify."*

**Author and co-authors: Indeed, stable isotope analyses were performed on the whole individual, minus its gut. We modified the sentence lines 236-237, page 8. It now reads:** *"In the case of intermediate-size specimens, the gut content was removed before stable isotope analyses."*.

*"Lines 265-266: sample volume as a proxy for habitat complexity. I am not entirely convinced by this, the long tubeworms seen at very low flow sites would provide a larger 'volume' by your definition, but their straight stiff tubes do not interlock in the same complex structure that intermediate flow tubes or even some high flow tubes do. I.e., shorter tubes may make for a smaller volume but it doesn't necessarily make for reduced habitat complexity. Southward and Tunnicliffe 1995, Tsurumi and Tunnicliffe (2003) & Tunnicliffe et al. 2014 discuss the phenotypic complexity of R. piscesae in terms of tube structure and the implications this has for faunal communities.*

**Author and co-authors: As reported above, in the light of the Reviewer's comments we improved our methodology to assess the structural complexity of tubeworm assemblages and consequently, to provide a better estimation of species densities, lines 220-227 page 8. We estimated the tubeworm surface area from measurements of the length and diameter of each tube. We added species densities (in ind m$^{-2}$) in Table 3. In the studies mentioned by this Reviewer, the authors compared the two known phenotypes of R. piscesae: long and skinny vs. short and fat. Each phenotype, that develop in different environments, is associated with different species and their comparison thus warrant more complex measures than tube length. However, in this study we only sampled the long and skinny phenotype.**

*Line 293: densities in m$^{-3}$. These numbers are incredibly large and speak to my point about tubeworm surface area… what does the number of individuals per m$^{-3}$ really mean for benthic organisms? Most of these species will be attached to substrata, i.e. the tubeworms, so that tubeworm surface area, measured in cm$^{-2}$ (as in Tsurumi and Tunnicliffe 2001) would probably be more relevant on a habitat scale than m$^{-3}$."*

**Author and co-authors: As mentioned earlier, we agree with the Reviewer and we improved our methodology to assess species densities by calculating the tubeworm surface area available for the fauna, as conducted by Tsurumi & Tunnicliffe (2001), Tsurumi and Tunnicliffe (2003) and Marcus et al. (2009); lines 220-227 page 8. However we believe that cm is a bit small as measurement unit for the habitat scale considering that several species are larger than 1 cm. We now present the developed surface (surface of tubeworms) and associated densities in m$^2$.**

*"Lines 286-299: macrofaunal densities. I presume these densities exclude R. piscesae abundance"*

**Author and co-authors: Yes, densities and richness exclude R. piscesae. This is now clarified at the beginning of the paragraph, lines 293-295 page 10:** *"A total of 148 005 individuals representing 35 macrofaunal species (excluding Ridgeia piscesae) were identified in the six tubeworm bushes (S1 to S6) sampled on the Grotto edifice."*

*"Line 315: acarida. Copidognathus papillatus is in the class 'Arachnida', subclass 'Acari'…. Family 'Halacaridae'. I'm not sure where 'acarida' comes from but it is not an accepted taxonomic name in the World Register of Marine Species (WORMS): http://www.marinespecies.org"*

**Author and co-authors: We changed "acarida" to "halacarid", lines 312-313 page 11:** *"[…] the halacarid Copidognathus papillatus, […]".*

*"Lines 350-351: "With a biomass ranging from…" In this and subsequent sentences, I presume you are referring to the biomass ranges across all samples at Grotto? Could you please clarify this in the text?"*

**Author and co-authors: As suggested by the Reviewer, we clarified the sentence, lines 344-345 page 12. It now reads:** *"With a total proportion of biomass ranging from 78.9 to 95.8 % (89.6 ± 6.8 %) across samples, […]".*

*"Lines 361-362: feeding guilds. See previous comment on how trophic guilds were assigned and applied. I am unclear here whether the discussion of biomass and guild relationships refers to guilds applied to taxa using the literature (including previous stable isotope studies) or whether the authors have re-assigned guilds from their own stable isotope data. A table with the species and guilds assigned using the literature, with an additional column indicating whether the present study agrees with these designations would be helpful."*

**Author and co-authors: As reported above, we provided a new table (Table 1) with the trophic guild and the nutritional mode of each species, as well as references.**

*"Line 390: "a pattern that has also…." I presume here you mean the pattern of dominance by a few species? This is worth clarifying, as it could be misread that the dominance of L. funcensis, D.globulus, A carldarei specifically is the pattern you are referring to. Consider breaking it into two sentences: "R. piscesae tubeworm assemblages sampled on the Grotto edifice were characterized by the dominance of a few species, such as L. fucensis, D. globulus and A. carldarei. Numerical dominance by a few species is a pattern that has also been observed from other hydrothermal sites on the world oceans….""*

**Author and co-authors: As suggested by the Reviewer, we modified the sentence lines 383-386 page 13. It now reads:** *"Tubeworm bushes sampled on the Grotto edifice were characterized by the dominance of a few species such as Lepetodrilus fucensis, Depressigyra globulus and Amphisamytha carldarei. Numerical dominance by a few species is a pattern that has also been reported from other hydrothermal sites of the world oceans […]".*

*"Lines 400-401: sample size. How much smaller were the samples from 2016, in terms of surface area/volume? How much material (as an estimate) was lost during collection?"*

**Author and co-authors: Sample size was indicated in Table 1, which is now referred to in the text. We don't see a loss of material during the collection samples and now specify it in the text, lines 395-397 page 13:** *"The areas sampled in 2016 were smaller than in 2015 and a problem with the sampling boxes may have led to the loss of some individuals, even though not visible from videos recorded by the submersible.".*

*"Line 412: what do you mean by "a high level of complexity?" a complex tubeworm bush structure? This needs some clarification."*

**Author and co-authors: We clarified this point on lines 406-408, page 14. Here is the new paragraph:** *"Tubeworm samples S1 and S3 were visually recognized as type V low-flow communities (Sarrazin et al., 1997), an advanced stage in the ecological succession characterized by a longer tube (18.5 ± 3.3 cm) and a higher level of structural complexity.".*

*"Lines 413-416. With the limited information you have on the character of your tubeworms from each sample, I'm not convinced you have sufficient evidence for this statement. You haven't assessed the nature of the biogenic structure for your samples (i.e. the physical*

*structure of your tubeworm bushes); you've calculated the potential volume of seawater your tubeworm bushes could occupy, which is not the same."*

**Author and co-authors: As suggested by this Reviewer, we modified species density measurements by calculating the total surface area of the tubeworm tubes under the assumption that they are cylinders, lines 220-227 page 8. These new data now provide the total surface available to the fauna for colonization, which is a proxy of the biogenic structure for this phenotype, and does support the role of engineering species in controlling diversity.**

*"Lines 429-430: why do your density ranges for A. carldarei exclude S4 and S6? If you're talking about the Grotto edifice as a whole, shouldn't these two samples be included?"*

**Author and co-authors: We agree with the Reviewer and modified the sentence lines 424-425, pages 14:** *"High densities of A. carldarei in R. piscesae tubeworm bushes (up to 93.4 %) […]".*

*"Line 430: High densities of what? A. carldarei? Please clarify in the text."*

**Author and co-authors: We modified the sentence line 424, page 14 by adding the name of the species. It now reads:** *"High densities of A. carldarei in R. piscesae tubeworm bushes […]".*

*"Lines 430-431: "…may be related to the specificity of this family with high ecological tolerance and, therefore…" Specificity and tolerance are opposite ends of a spectrum, consider rephrasing to "….may be related to this ampharetid's tolerance to environmental conditions and, therefore….""*

**Author and co-authors: As suggested by the reviewer, we modified the sentence on lines 424-427 page 14. It now reads:** *"High densities of A. carldarei in R. piscesae tubeworm bushes (up to 93.4 %) may be related to this ampharetid's tolerance to environmental conditions and, therefore, to their ability to take advantage of a wide range of ecological niches (McHugh and Tunnicliffe, 1994).".*

*"Lines 441-442: "The grazing of new recruits may also limit species diversity" Do you mean the grazing action of new recruits to the community limits species diversity, or the grazing action of the community on new recruits limits species diversity? Please clarify."*

**Author and co-authors: The grazing pressure comes from the action of the community (mainly gastropods) on new recruits. We clarified this, lines 433-436 page 15:** *"We hypothesize that the numerical dominance of gastropods negatively affected species diversity by monopolizing space and nutritional resources and by potentially grazing new recruits, therefore reducing the settlement of other vent species."*.

*"Lines 455-457: three pools of isotopically distinct microbial production. Do you think L. fucensis really feeds on its own separate pool of microbial production? Or does its contrasting isotopic signature reflect the fact that it may be utilising multiple microbial sources at any one time? As you mention later (lines 569-575), it can host bacteria on its gill filaments and eat them, along with grazing and suspension feeding activities. It might be worth introducing this concept here instead of the brief reference to Bates (2007) at line 485."*

**Author and co-authors: We agree that the central position of L. fucensis could be linked to its ability to use multiple microbial sources. We changed 'three large pools' by 'a wide range' to highlight the existence of different pools of microbial sources without restraining each species to a given pool, lines 451-454 page 15:** *"In the present study, the contrasting isotope compositions of the gastropods Provanna variabilis, Lepetodrilus fucensis and the polychaete R. piscesae suggest a wide range of isotopically distinct, symbiotic and/or free-living microbial production available to primary consumers."*.

*"Lines 461-462: "…species deriving their food sources from siboglinid tubeworms are rare." I don't quite agree here – there are multiple species that do eat R. piscesae, including B tunnicliffae as you mention at line 469. Buccinum thermophilum also preys on R. piscesae. That the isotopic signatures don't closely match these predators to their prey doesn't mean that the predators are rare; instead it suggests the predators are not specialist, i.e. they eat more than just R. piscesae."*

**Author and co-authors: We agree with the Reviewer and modified the sentence on lines 456-458 page 15:** *"Despite that R. piscesae contributed to 86 % of the total biomass, a low*

*number of species displayed similar δ¹³C values, suggesting that specialist species deriving the majority of their food sources from siboglinid tubeworms are rare."*.

*"Line 471: "Predation on tubeworms was confirmed…" Predation by whom? There are many images/video clips of B. tunnicliffae predating on R. piscesae; do you have imagery of L. piscesae doing the same? This sentence needs some clarification."*

**Author and co-authors:** **We clarified the sentence on lines 467-468, page 16:** *"Moreover, the predation of tubeworms by polynoids is often observed as shown here on a video sequence […]"*. **The resolution of the video did not enable us to identify with certainty the polynoids to the species level.**

*"Lines 480-482: "most vent species display an isotopic composition centred on L. fucensis" What do you mean by this? That most of the species have an isotopic composition similar to L. fucensis? Please clarify."*

**Author and co-authors:** **We clarified the sentence, lines 477-478 page 16. It now reads:** *"Food webs obtained in this study revealed that most vent species display a δ¹³C similar to L. fucensis, but with slightly higher δ¹⁵N values."*.

*"Lines 483-487: the role of L. fucensis. Do you think that the wide-ranging feeding strategies of this limpet really exert pressure on the availability of resources for others? Or does the range of strategies open up more opportunities for L. fucensis, allowing them to take advantage of resources that others cannot? See you text at lines 569-575."*

**Author and co-authors:** **These are two interrelated and valid explanations. By using a wide range of food sources, which is possible thanks to their capacity to use three different feeding modes, not only L. fucensis benefits from more resource availability, but it also reduces the resource availability for other taxa. The sentence was modified to reflect this lines 480-484 page 16:** *"The wide-ranging feeding strategies of this limpet may exert a high pressure on the availability of nutritional resources for others vent species. Likewise, the range of feeding strategies open up more opportunities for L. fucensis, allowing this limpet to take advantage of resources that others species cannot."*.

*"Lines 487-490: Paralvinella species. These line are somewhat unclear, are all Paralvinella selective deposit feeders? Was the stable isotope composition more variable in terms of intraspecific variation or in terms of interspecific variation among samples?"*

**Author and co-authors: In our study, all Paralvinella species are suspension and/or deposit feeders [1,2,3]. The original sentence was incorrect, we improved it for greater clarify, lines 484-491, page 16:** *"The four Paralvinella species observed in our samples, which are suspension and/or deposit-feeders (Desbruyères and Laubier, 1986; Tunnicliffe et al., 1993), displayed low or negative $\delta^{15}N$ values. These lowest $\delta^{15}N$ values may be related to the nutrition of a microbial pool based on local nitrogen sources. In fact, the ammonium produced during the microbial degradation of organic matter appears to be usually $^{15}N$-depleted (Lee and Childress, 1996). Amongst these Paralvinella species, Paralvinella sulfincola, Paralvinella palmiformis and Paralvinella dela shared the same isotopic niche while Paralvinella pandorae displays a distinct isotopic composition."*

[1]**Desbruyères, D., & Laubier, L. (1986). Les Alvinellidae, une famille nouvelle d'annélides polychètes inféodées aux sources hydrothermales sous-marines: systématique, biologie et écologie. Canadian Journal of Zoology, 64(10), 2227-2245.**

[2]**Desbruyeres, D., & Laubier, L. (1991). Systematics, phylogeny, ecology and distribution of the Alvinellidae (Polychaeta) from deep-sea hydrothermal vents. Ophelia, 5(Suppl), 31-45.**

[3]**Grelon, D., Morineaux, M., Desrosiers, G., & Juniper, S. K. (2006). Feeding and territorial behavior of Paralvinella sulfincola, a polychaete worm at deep-sea hydrothermal vents of the Northeast Pacific Ocean. Journal of experimental marine biology and ecology, 329(2), 174-186.**

*"Line 494: "Grotto primary consumers were dominated by grazers and deposit feeders" Where is your evidence/data for this? A table of your trophic guild designations would be really useful to refer to here."*

**Author and co-authors: From the original description and the literature, we have seen that Grotto primary consumers were dominated by grazers and deposit feeders. Our isotopic results are consistent with these observations. As suggested by the reviewer, we added the Table 1, in which the trophic guild and the nutritional mode of each species can be found as well as references.**

*"Lines 500-501: P. sulfincola low nitrogen values. This sentence needs some clarification; the feeding behaviour of P sulfincola can only explain the low nitrogen values for that species, not all Paralvinella species, as they may (and in fact do) display different feeding strategies."*

**Author and co-authors: We decided to remove the following sentence:** *"The polychaete P. sulfincola can feed directly on microbial biofilms on the substratum around its tube opening (Grelon et al., 2006), which may explain the low $\delta^{15}N$ values of alvinellids in the present study."* **to maintain this part of the discussion general to the Paralvinella genus.**

*"Line 508: "the comparatively small size of P. pandorae" How large are the P. pandorae in your samples compared to the size range of your other Paralvinella species?"*

**Author and co-authors: We did not specifically measure the individuals of Paralvinella species. Individuals of P. pandorae were within their range size. It is known from the literature and the original descriptions of Paralvinella species that P. pandorae is significantly smaller than the other species. Therefore, we removed "(Lelièvre Y. , personal observations)" to cite instead the relevant literature. Indeed, as reported by Desbruyères and Laubier (1986; original description) and Tunnicliffe et al. (1993; original description), P. pandorae is no more than 2 cm in length while P. palmiformis and P. sulfincola can be up to 6 cm and 8 cm in length, respectively. This difference in size is not specific to this study but occurs at all study sites on the Juan de Fuca ridge. We added these references in the sentence, lines 506-509 page 17:** *"The comparatively small size of P. pandorae compared with other alvinellid species (Desbruyères and Laubier, 1986; Tunnicliffe et al., 1993) may be linked to the presence of interspecific competition for food resources and/or a diet based on an isotopically distinct microbial source.".*

*"Lines 517-520: wide range of carbon values. Does this mean the carbon values provide less supporting evidence for your designated guilds than the nitrogen values? As phrased, it appears that you are choosing to go with the nitrogen data as it better fits your guild concepts, despite the range in carbon... Please could you clarify? Again, having a table with your guild designations for each species, whether the carbon and nitrogen values from your study support these, and even how your values compare to other studies (e.g. Bergquist et al. 2007) would be really helpful. See Table 1 in Bergquist et al. 2007 for a starting point."*

*Author and co-authors: The term "guild" was formally proposed as "a group of species that exploit the same class of environmental resources in a similar way" (Root, 1967)[1]. Therefore, a trophic guild is defined by the nature of the prey (predator, detritivore, bacterivore, etc.). But species of the same trophic guild may have different kinds of nutritional modes, e.g. bacterivore with surface deposit feeding or grazing. As indicated in the introduction section lines 113-120 pages 4-5, the carbon isotope composition ($\delta^{13}$C) reflects basal resources while nitrogen isotope composition ($\delta^{15}$N) provides information on trophic levels. The wide range of carbon isotopic composition provides additional information on resource partitioning within the predator feeding guild and is not in contradiction with the fact that predator consistently display higher $\delta^{15}$N. However, as previously reported, we added the Table S1, in which the nutritional modes of each species and references can be found.*

*[1]Root, R. B. (1967). The niche exploitation pattern of the blue‑gray gnatcatcher. Ecological monographs, 37(4), 317-350.*

"Lines 521-529: These lines are unclear with respect to L. kincaidi. The high nitrogen values make it a "top predator", but isotopic variability suggests it has "highly diversified food resources". So it isn't just a predator but also exhibits other feeding strategies? Please provide some clarification here, again a table of isotopic values by species and your guild designations (cf Bergquist et al. 2007) would be really useful for interpretation."

*Author and co-authors: Being a predator does not prevent to have a highly diversified food resource. Also, we removed 'top' in 'top predator' because our data does not provide evidence that higher predators (crabs, fish) are absent. Actually, crabs and fish were observed within the study area, we added a sentence lines 524-527 pages 17-18: "The presence of Zoarcidae eelpouts Pachycara gymninium and Oregoniidae spider crabs Macroregonia macrochira on the Grotto edifice, not sampled but observed in the video recorded by the TEMPO-mini ecological module, could also played a role of predator within the ecosystem." As mentioned above, we added Table 1, in which the trophic guild and the nutritional mode of each species can be found, with references.*

*"Lines 539-541: How did your results overall compare to Bergquist et al. 2007? Did you get similar values and ranges for the same species? It would be helpful to compare your work to theirs here."*

**Author and co-authors: The isotopic composition of species have shown similar results to those obtained by Bergquist et al. (2007) with, however, less variation in isotope carbon and nitrogen. As suggested by the Reviewer, we added a sentence on lines 445-447 page 15:** *"In this study, the position of species in the food webs (trophic structure) was consistent with the ones reported in Bergquist et al. (2007), with however, less variability in carbon and nitrogen stable isotope."*.

*"Lines 561-566: gastropods. The group gastropoda is very large and the species you have; D. globulus, P. variabilis & L. fucensis come from completely different families, they are even in different sub-classes (Neomphalina, Caenogastropoda, Vetigastropoda respectively) so I'm not convinced the concept of "closely related" (line 560) in relation to these species and resource portioning is reasonable. They are not closely related; it is not so surprising that they have different feeding strategies and occupy different niches."*

**Author and co-authors: We agree with the Reviewer, this is true for the polychaetes as well. The sentences were modified to make it clear to readers that we do not consider these gastropod species as closely related, lines 561-562 page 18:** *"[…] food partitioning may occur between different species of the same or related taxonomic family […]"*.

*"Lines 588-589: B. thermophilum carbon signatures. This species has been observed predating on tube worms, could the higher carbon values support this and/or other predatory behaviours?"*

**Author and co-authors: The carbon values do not provide information on the nutritional modes of species. In general, the isotopic composition of B. thermophilum is not incompatible with a diet including R. piscesae, although does not allow to confirm this. However, like Bergquist et al. (2007) on Buccinum viridum, our results suggest a scavenger/detritivore behaviour for B. thermophilum.**

*"Lines 596-598 & 611-613: I still have concerns that your characterisation of the tubeworm habitat is not sufficient to make this statement, see previous comments relating to tubeworm plasticity and habitat volume as a proxy."*

**_Author and co-authors:_** **We believe that the new method to estimate the surface of the 3-d biogenic structure (tubeworm surface area available) now provides a good proxy of habitat complexity. Also, we would like to remind that our data showed a significant correlation between tube length and species diversity.**

**III. Technical corrections**

**All minor technical corrections were taken into account.**

*"Line 35 & 36: would read better if gastropods are listed first. See previous sentence."*

**_Author and co-authors:_** **We modified the sentence, lines 36-37 page 2.**

*"Lines 88 – 98: multiple occurrences of 'e.g.' and 'etc.'; could these sentences be more specific?"*

**_Author and co-authors:_** **We removed the "etc.", lines 92-94 page 4. We used "e.g" to avoid a sentence too long.**

*"Line 172: Middle Valley isn't really a site per se; it's more of a seabed area and there are sites within it. Might just read better as 'Middle Valley' without 'site'."*

**_Author and co-authors:_** **We modified the sentence, line 170 page 6.**

*"Lines 174-175: would read better as "The five major vent fields – Sasquatch, Salty Dawg, High Rise, Main Endeavour Field (MEF) and Mothra – found along the Endeavour axial valley are each separated by 2-3 km." Please note that MEF stands for Main Endeavour Field, not just Main Endeavour."*

**_Author and co-authors:_** **We modified the sentence, lines 172-174 page 6.**

*"Line 183: "a relative stability in years": would read better as "but is relatively stable across years"*

*Author and co-authors: We modified the sentence, lines 184-185 page 7.*

*"Line 234: "About 1.3-1.4 mg of the powder was precisely measured" consider substituting 'about' for 'approximately'."*

*Author and co-authors: As suggested by the reviewer, we substituted "about" for "approximately", line 240 page 8.*

*"Line 268: Table 1. Where is Table 1? I can't find it in the pdf. I also cannot find Table 2!"*

*Author and co-authors: As previously indicated, the two tables of this paper were prepared correctly, but it seems that a problem might have occurred during the submission/revision process.*

*"Lines 279-280: the lowest diversity values. This sentence would read better if the lowest diversity values were listed first, in the order of increasing diversity values."*

*Author and co-authors: We modified this paragraph, lines 283-286 page 10.*

*"Lines 301, 329 & 337: "more specifically". You might consider an alternative start to these sentences, such as "In more detail"*

*Author and co-authors: As suggested by the referee, we changed the beginning of the sentences on line 322 page 11 and line 330 page 11.*

*"Line 323: "…S6 was also dominated L. fucensis" should read "…S6 was ably dominated by L. fucensis"*

*Author and co-authors: We think that "ably" is not appropriate in this sentence so we kept is as it was.*

*"Line 348: "engineer polychaete" should probably read "ecosystem engineering polychaete""*

*Author and co-authors: The term "engineer polychaete" was replaced by "ecosystem engineering polychaete", line 341 page 12.*

*"Lines 372-376. Consider rephrasing slightly to make it clear the reference is for all the species richness records. Alternative phrasing could put the reference at the start of the*

*sentence: "Tsurumi and Tunnicliffe (2003) reported the presence of 39 macrofaunal species…. 19 species in 2 collections from the CoAxial Segment (JdFR).""*

**Author and co-authors:** **We modified the sentence, lines 366-367 page 12.**

*"Line 379: "engineer mussel beds: should probably read "ecosystem engineering mussel beds""*

**Author and co-authors:** **We modified the sentence, line 373 page 13.**

*"Line 382: "worldwide hydrothermal ecosystems". In the context of this sentence, I think you mean biogeographic regions or provinces? Consider changing to "Faunal dissimilarities between biogeographic regions…"""*

**Author and co-authors:** **We modified the sentence, line 377 page 13.**

*"Lines 442-445: this sentence is to me unclear. Please consider re-phrasing/clarifying. Also note repeat Sarrazin et al. (2002) reference.""*

**Author and co-authors:** **We modified the sentence, lines 436-439 page 15.**

---

## Author Comment (AC2) · 23 Feb 2018

We are grateful to the reviewer for the useful and relevant suggestions that helped us significantly improve the manuscript. We have dealt with all the comments/questions following his/her suggestions.

Please fin attached our response to referee's comments.

Please also note the supplement to this comment: https://www.biogeosciences-discuss.net/bg-2017-411/bg-2017-411-AC2-supplement.pdf

[Figure]

**Supplement:**

**Responses to Editor and Referees**

**Anonymous Referee**

"Biodiversity and trophic ecology of hydrothermal vent fauna associated with tubeworm assemblages on the Juan de Fuca Ridge"

**bg-2017-411**

**I. General comments**

"The manuscript by Lelièvre and colleagues presents interesting information on the structure, biodiversity, and trophic structure of hydrothermal vent sites on the Main Endeavour Field on the Juan De Fuca ridge. This manuscript and their data take a nice holistic approach and compare a variety of different communities that they suggest are different successional stages based on observations there, thus link successional patterns to the greater community and trophic structure that live there. The manuscript is well crafted and they found that predator-prey relationships were not as dominant as the important role of ecosystem engineers in structuring the communities. The research is also important as it provides a nice baseline for future studies that work at these locations, which are near a cabled array system so this is likely to be highly referenced site in the future. My only complaint is a slightly cursory treatment of the isotopic data and the use of individuals per m3 instead of m2 for benthic communities which I believe are muddling some of the results. This is a nice manuscript that advances the field."

Author and co-authors: We are grateful to the reviewer for the useful and relevant suggestions that helped us significantly improve the manuscript. We have dealt with all the comments/questions following his/her suggestions. Regarding the treatment of the isotopic data, the use of more complex indexes such Layman et al. (2007) or Cucherousset & Villéger (2015) were tested here on our data but did not provide more insight on the observed patterns. We thus limited the complexity of the analyses for clarity. Also, this reviewer notes in a comment below 'I am not sure that a more in depth analysis would be possible with the heterogeneous and every shifting baseline caused by the diversity of microbial communities so that is not what I am recommending [...]'. We thus believe that extra analyses would have only complexified the paper without adding any relevant information. Regarding the density of individuals, because the other reviewer also suggested calculating density by surface of tubeworm, we now provide the information in the manuscript (see details below). Our responses are given in bold under each comment. We hope that our comments and modifications increased the quality of our paper for a publication in Biogeosciences.

**II.** Reviewer comments**

"I would say that the introduction could use more specifics about the stable isotopes at vents where there can be pretty significant variation in both C and N at the base of the food web due to symbionts (often negative N) in contrast to other inputs, plus the relatively high N of phytodetritus. It adds a dimension to the isotopic analyses in other systems and without this mention may confuse the reader until it is discussed in the discussion. Simply a sentence or two in the introduction could help the readers have a better foundation for this. L511 in the discussion does point towards it but without specific examples. Simply adding a sentence at line 119 saying that these different sources of primary production vary in del 15 N making clear trophic analysis more complex would be one way to do that."

Author and co-authors: As suggested by the Reviewer, we added a few sentences about this subject, lines 126-134 page 5 of the manuscript: "The carbon signature ( $\delta^{13}$ C) of primary producers differs according to their carbon fixation pathways that differentially fractionate inorganic carbon sources. Despite the fact that the nitrogen signature ( $\delta^{15}$ N) does not discriminate primary producers, the variability of  $\delta^{15}$ N signatures can be associated to their origins and, also, to local biogeochemical processes (Bourbonnais et al., 2012; Portail et al., 2016). Moreover, due to its degradation in the water column, photosynthesis-derived organic matter is characterized by high  $\delta^{15}$ N values in comparison with local vent microbial producers, which are associated with low or negative values characteristic of local inorganic nitrogen sources (Conway et al., 1994)."

"Results – I am torn on the use of the 3-dimensional space for extrapolating up the total density of fauna. I believe the numbers could be important but really it is a two-d surface area that is expanded up by ecosystem engineers but limited by the energy input and space, which is more 2-d. At a minimum, a statement and comparison of the 2-d abundance would

be an important comparison, especially when comparing different habitats where the increased area of tube worms will decrease the extrapolation up (i.e. fewer fauna per  $m^2$  but with a lower height measurement will come up with a much higher ind. M-3 value than one that had a higher density on the per m2 but since you measured a larger area will be fewer on the m3 metric)."

Author and co-authors: We do not agree that height measurement will bias the density of organisms. Macrofauna species inhabit the full volume of worms with gastropods grazing along the entire length of the tubes. This space exploitation is visible from the video imagery and on the tubes when brought back to the surface. However, in the light of the comments provided by both reviewers, we improved our methodology to assess the structural complexity of tubeworm assemblages using  $m^2$ , lines 220-227 page 8 of the manuscript. To estimate the tube surface area, we measured the tubes both lengthwise and crosswise (i.e. diameter). By assuming the tube has a cylinder shape, we obtained the developed surface (surface of tubeworms) in  $m^2$ . We therefore added species densities (ind  $m^{-2}$ ) in Table 3. The strong correlation between the developed surface areas and species abundances further supports the fact that macrofauna colonize the entire surface available in a 3-dimensional space.

"I found the treatment of the isotope data not comprehensive enough to support the conclusions made. Specifically, L 362 identifies a shift from bactivorous to predator guild, but which species belong to which? How is this shown by these data? I also question whether the term "trophic network" is appropriate. Really these data are just presented and then scaled by biomass, which I like, but is not a trophic network per se. Either modify the term or expand the analyses performed to look more at connections. I am not sure that a more in depth analysis would be possible with the heterogeneous and every shifting baseline caused by the diversity of microbial communities so that is not what I am recommending, but instead I would avoid the term trophic network."

Author and co-authors: We agree with the Reviewer that our isotopic data were not comprehensive enough to support the statement found on line 362 (previous manuscript version). We removed this sentence. As suggested, the expression "trophic network" was replaced with "food/trophic web". "Figures 1-3 are very nice. Figures 4 and 5 have too small of font on the axis and the grey background clutters the visuals, especially when numbered, also too small. I do not consider these two figures ready for publication. The grey should be removed, the lines within the text should be removed and ideally a key with the colors and the species should be included so the reader is not forced to delve heavily into the figure legend to know what they are looking at." Author and co-authors: As suggested by the Reviewer, we modified Figures 4 and 5 so they are now suitable for publication in Biogeosciences (see below).

**III. Small suggestions**

"L37 "Fairly" long tubes comes across as vague. Since the actual values are known, please just include them."

Author and co-authors: We modified the sentence and added the real measurements line 38 page 2 of the manuscript.

"L122 – I would suggest adding in "average rate of +3.2" as that is a mean of multiple, often highly variable values."

Author and co-authors: As suggested by the Reviewer, we modified the sentence on line 119 page 5 of the manuscript.

"L144 – The sentence that starts on this line seems out of place in this paragraph. It should be either removed or rephrased as to why this builds upon what was said before."

Author and co-authors: As suggested by the Reviewer, the sentence has been removed. The entire paragraph was actually reworked and shortened following additional comments from the other reviewer.

"L294-296 It seems that these should either be reported in dm3 or with a different number of significant figures as there was not a m3 counted. I understand why m3 was used so suggest just 17 x 106 etc. but also comparing them in a m-2 context."

Author and co-authors: As strongly suggested by both reviewers, we improved species density measurements by calculating the tubeworm surface area. Species density is now expressed in number of individuals per square meter of tubeworm surface (ind m-2).

*"L290- again question whether the right number of significant digits is used on the percentage Line 361 "contributed – 16.4%…""*

Author and co-authors: All percentages have been harmonised and expressed with one significant digit.

"L386 – also sampling approaches. Any of the sampling that has occurred with a mussel pot or a Bushmaster could also lead to differences in diversity simply due to methodology. In addition, not suctioning the area could also lead to lesser diversity in other studies."

Author and co-authors: We agree with the Reviewer and modified the sentence, lines 375-376 page 13 of the manuscript: "Variation between sites and regions may be related to discrepancies in sampling effort and methodologies.".

*"L414 – I question whether a trophic network is the right word here considering the analyses done."*

Author and co-authors: As previously reported, the term "trophic network" was replaced with "food web" throughout the manuscript.

**Figure 4.** Stable isotope bi-plots showing vent consumers' isotope signatures (mean  $\delta^{13}$ C versus  $\delta^{15}$ N values ± standard deviation) for the six vent assemblages sampled on the Grotto hydrothermal edifice. Each vent species is designated by a number: 1 = Ridgeia piscesae; 2 = Provanna variabilis; 3 = Depressigyra globulus; 4 = Lepetodrilus fucensis; 5 = Buccinum thermophilum; 6 = Clypeosectus curvus; 7 = Amphisamytha carldarei; 8 = Branchinotogluma tunnicliffeae; 9 = Lepidonotopodium piscesae; 10 = Levensteiniella kincaidi; 11 = Nicomache venticola; 12 = Paralvinella sulfincola; 13 = Paralvinella palmiformis; 14 = Paralvinella pandorae; 15 = Paralvinella dela; 16 = Hesiospina sp. nov.; 17 = Sphaerosyllis ridgensis; 18 = Ophryotrocha globopalpata; 19 = Berkeleyia sp. nov.; 20 = Protomystides verenae; 21 = Sericosura sp.; 22 = Euphilomedes climax; 23 = Xylocythere sp. nov.; 24 = Copepoda; 25 = Copidognathus papillatus; 26 = Paralicella cf. vaporalis; 27 = Helicoradomenia juani. Known trophic guilds are distinguished by a colour code: pink: symbiont; green: bacterivores; blue: scavengers/detritivores; red: predators. For more information on the interpretation of guilds, please consult the web version of this paper.

---

## Author Comment (AC3) · 23 Feb 2018

**Table 1.** Trophic guild and nutritional modes of macrofaunal species associated with the *Ridgeia piscesae* tubeworm assemblages of the Grotto hydrothermal edifice (Main Endeavour Field, Juan de Fuca Ridge). An asterisk (*) marks the original description of the species.

| Species | Trophic guil - Nutritional mode | Reference(s) |
|---|---|---|
| **Annelida** | | |
| **Polychaeta** | | |
| Siboglinidae | | |
| *Ridgeia piscesae* | Symbiotic | Jones,1985*; Southward et al., 1995; Bergquist et al., 2007; This study |
| Maldanidae | | |
| *Nicomache venticola* | Bacterivore - Surface deposit feeder or grazer | Blake and Hilbig, 1990*; Bergquist et al., 2007; This study |
| Dorvilleidae | | |
| *Ophryotrocha globopalpata* | Predator | Blake and Hilbig, 1990*; Bergquist et al., 2007; This study |
| Orbiniidae | | |
| *Berkeleyia* sp. nov. | Scavenger/detritivore - Suspension feeder | Jumars et al., 2015; This study |
| Hesionidae | | |
| *Hesiospina* sp. nov.[1] | Predator | Bonifácio et al., *in preparation*\*; This study |
| Phyllodocidae | | |
| *Protomystides verenae* | Predator | Blake and Hilbig, 1990*; Bergquist et al., 2007; This study |
| Polynoidae | | |
| *Branchinotogluma tunnicliffeae* | Predator | Pettibone, 1988*; Bergquist et al., 2007; This study |
| *Branchinotogluma* sp. | Predator | - |
| *Lepidonotopodium piscesae* | Predator | Pettibone, 1988*; Levesque et al., 2006; Bergquist et al., 2007; This study |
| *Levensteiniella kincaidi* | Predator | Pettibone, 1985*; Bergquist et al., 2007; This study |
| Sigalionidae | | |
| *Pholoe courtneyae* | Predator | Blake, 1995*; Sweetman et al., 2013 |
| Syllidae | | |
| *Sphaerosyllis ridgensis* | Predator | Blake and Hilbig, 1990*; Bergquist et al., 2007; This study |
| Alvinellidae | | |
| *Paralvinella dela* | Bacterivore - Surface deposit feeder or grazer; suspension feeder | Detinova et al., 1988*; This study |
| *Paralvinella palmiformis* | Bacterivore - Surface deposit feeder or grazer; suspension feeder | Desbruyères and Laubier, 1986*; Desbruyères and Laubier, 1991; Levesque et al., 2003; This study |
| *Paralvinella pandorae* | Bacterivore - Surface deposit feeder or grazer; suspension feeder | Desbruyères and Laubier, 1986*; Desbruyères and Laubier, 1991; Levesque et al., 2003; This study |
| *Paralvinella sulfincola* | Bacterivore - Surface deposit feeder or grazer; suspension feeder | Tunnicliffe et al., 1993*; Levesque et al., 2003; This study |
| Ampharetidae | | |
| *Amphisamytha carldarei* | Scavenger/detritivore - Surface deposit feeder or grazer | Stiller et al., 2013*; McHugh and Tunnicliffe, 1994; Bergquist et al., 2007; This study |
| Ctenodrilidae | | |
| *Raricirrus* sp. | Bacterivore - Surface deposit feeder or grazer | Jumars et al., 2015 |
| Spionidae | | |
| *Prionospio* sp. | Bacterivore - Surface deposit feeder or grazer | Jumars et al., 2015 |
| **Mollusca** | | |
| **Aplacophora** | | |
| Simrothiellidae | | |
| *Helicoradomenia juani* | Predator | Scheltema and Kuzirian, 1991*; Bergquist et al., 2007; This study |
| **Gastropoda** | | |
| Buccinidae | | |
| *Buccinum thermophilum* | Scavenger/detritivore - Surface deposit feeder or grazer | Harasewych and Kantor, 2002*; Martell et al., 2002; This study |
| Provannidae | | |
| *Provanna variabilis* | Bacterivore - Surface deposit feeder or grazer | Warén and Bouchet, 1986*; Bergquist et al., 2007; This study |
| Peltospiridae | | |
| *Depressigyra globulus* | Bacterivore - Surface deposit feeder or grazer | Warén and Bouchet, 1989*; Bergquist et al., 2007; This study |
| Clypeosectidae | | |
| *Clypeosectus curvus* | Predator | McLean, 1989*; Bergquist et al., 2007; This study |
| Lepetodrilidae | | |
| *Lepetodrilus fucensis* | Symbiotic; Bacterivore - Surface deposit feeder or grazer; suspension feeder | McLean, 1988*; Fox et al., 2002; Bates et al., 2007; Bergquist et al., 2007; This study |
| **Arthropoda** | | |
| **Arachnida** | | |
| Halacaridae | | |
| *Copidognathus papillatus* | Predator | Krantz, 1982*; Bergquist et al., 2007; This study |
| **Amphipoda** | | |
| Alicellidae | | |
| *Paralicella cf. vaporalis* | - | Barnard and Ingram, 1990* |
| Calliopiidae | | |
| *Oradarea cf. walkeri* | - | Shoemaker, 1930* |
| *Letpamphopus* sp. | - | - |
| **Crustacea** | | |
| Philomedidae | | |
| *Euphilomedes climax* | Bacterivore - Surface deposit feeder or grazer | Kornicker, 1991*; This study |
| Cytherudidae | | |
| *Xylocythere* sp. nov.[2] | Bacterivore - Surface deposit feeder or grazer | Tanaka et al., *in preparation*\*; Maddocks and Steinceck, 1987; This study |
| **Pycnogonida** | | |
| Ammotheidae | | |
| *Sericosura verenae* | Bacterivore - Surface deposit feeder or grazer | Child, 1987*; Bergquist et al., 2007; This study |
| *Sericosura venticola* | Bacterivore - Surface deposit feeder or grazer | Child, 1987*; Bergquist et al., 2007; This study |
| *Sericosura cf. dissita* | Bacterivore - Surface deposit feeder or grazer | Child, 2000*; This study |
| **Nemertea** | | |
| **Unidentified** | Predator | - |
| **Echinodermata** | | |
| **Ophiuroidae** | - | - |

[1]This species refers to *Hesiospina legendrei* that is currently under description (Bonifácio et al., *in preparation*)
[2]This species refers to *Xylocythere sarrazinae* that is currently under description (Tanaka et al., *in preparation*)

Table 2. Percentage abundance x 100 (% Ab.), faunal density (D. ind m⁻²), volume (V. ind m⁻²) and relative biomass x 100 (% Biom.) of the different macrofaunal taxa (>250 μm) identified in the 6 sampling units (S1 to S6) on the Grotto edifice. The taxa were identified to the lowest possible taxonomic level.

| Species | S1 - Assemblage V low-flow | | | | S2 - Assemblage V low-flow | | | | S3 - Assemblage V low-flow | | | | S4 - Assemblage IV | | | | S5 - Assemblage V low-flow | | | | S6 - Assemblage III | | | |
|---|---|---|---|---|---|---|---|---|---|---|---|---|---|---|---|---|---|---|---|---|---|---|---|---|
| | % Ab. | D. (ind m⁻²) | V. (ind m⁻²) | % Biom. | % Ab. | D. (ind m⁻²) | V. (ind m⁻²) | % Biom. | % Ab. | D. (ind m⁻²) | V. (ind m⁻²) | % Biom. | % Ab. | D. (ind m⁻²) | V. (ind m⁻²) | % Biom. | % Ab. | D. (ind m⁻²) | V. (ind m⁻²) | % Biom. | % Ab. | D. (ind m⁻²) | V. (ind m⁻²) | % Biom. |
| **Annelida** | | | | | | | | | | | | | | | | | | | | | | | | |
| **Polychaeta** | | | | | | | | | | | | | | | | | | | | | | | | |
| Maldanidae | | | | | | | | | | | | | | | | | | | | | | | | |
| *Nicomache venticola* | 0.1 | 8 | 1604.5 | 0.2 | 0.1 | 14.1 | 7033.6 | 0.1 | 0.2 | 35.5 | 6363.3 | 1.1 | 0 | 0 | 0 | 0 | 0 | 0 | 0 | 0 | 0 | 0 | 0 | 0 |
| Dorvilleidae | | | | | | | | | | | | | | | | | | | | | | | | |
| *Ophryotrocha globopalpata* | 0.1 | 4.2 | 849.5 | <0.1 | 0.1 | 9.8 | 4893 | <0.1 | 1.3 | 202 | 36241.1 | <0.1 | <0.1 | 1.8 | 934.6 | <0.1 | <0.1 | 10 | 1851.9 | <0.1 | 0 | 0 | 0 | 0 |
| Orbiniidae | | | | | | | | | | | | | | | | | | | | | | | | |
| *Berkeleyia* sp. nov. | 0 | 0 | 0 | 0 | 0 | 0 | 0 | 0 | <0.1 | 6.3 | 1137.8 | <0.1 | 0 | 0 | 0 | 0 | 0 | 0 | 0 | 0 | 0 | 0 | 0 | 0 |
| Hesionidae | | | | | | | | | | | | | | | | | | | | | | | | |
| *Hesiospina* sp. nov. | 0.1 | 9.8 | 1982.1 | <0.1 | 0.1 | 10.4 | 5198.8 | <0.1 | 0.1 | 14.3 | 2570.6 | <0.1 | 0 | 0 | 0 | 0 | 0 | 0 | 0 | 0 | 0 | 0 | 0 | 0 |
| Phyllodocidae | | | | | | | | | | | | | | | | | | | | | | | | |
| *Protomystides verenae* | <0.1 | 0.5 | 94.4 | <0.1 | <0.1 | 0.6 | 305.8 | <0.1 | <0.1 | 0.2 | 42.1 | <0.1 | 0 | 0 | 0 | 0 | 0 | 0 | 0 | 0 | 0 | 0 | 0 | 0 |
| Polynoidae | | | | | | | | | | | | | | | | | | | | | | | | |
| *Branchinotogluma tunnicliffeae* | <0.1 | 0.7 | 141.6 | <0.1 | 0.1 | 9.2 | 4587.2 | <0.1 | <0.1 | 2.6 | 463.6 | <0.1 | 0.2 | 47.4 | 25233.7 | 0.2 | <0.1 | 20 | 3703.7 | <0.1 | 0.3 | 16.4 | 400 | <0.1 |
| *Branchinotogluma* sp. | 0 | 0 | 0 | 0 | 0 | 0 | 0 | 0 | 0 | 0 | 0 | 0 | <0.1 | 1.8 | 934.6 | <0.1 | 0 | 0 | 0 | 0 | 0 | 0 | 0 | 0 |
| *Lepidonotopodium piscesae* | 0 | 0 | 0 | 0 | <0.1 | 1.2 | 611.6 | 0 | <0.1 | 3.8 | 674.3 | <0.1 | 0.2 | 43.9 | 23364.5 | 0.9 | 0.1 | 50.1 | 9259.3 | 0.2 | 0.2 | 10.9 | 266.7 | 0.1 |
| *Levensteiniella kincaidi* | 0.1 | 8.2 | 1651.7 | 0.1 | 0.1 | 10.4 | 5198.8 | <0.1 | <0.1 | 4 | 716.4 | <0.1 | <0.1 | 3.5 | 1869.2 | <0.1 | <0.1 | 20 | 3703.7 | <0.1 | 0 | 0 | 0 | 0 |
| Sigalionidae | | | | | | | | | | | | | | | | | | | | | | | | |
| *Phobe courtneyae* | <0.1 | 0.2 | 47.2 | <0.1 | 0 | 0 | 0 | 0 | 0 | 0 | 0 | 0 | 0 | 0 | 0 | 0 | 0 | 0 | 0 | 0 | 0 | 0 | 0 | 0 |
| Syllidae | | | | | | | | | | | | | | | | | | | | | | | | |
| *Sphaerosyllis ridgensis* | 3.6 | 257.6 | 51911.3 | <0.1 | 1 | 170.2 | 85015.3 | <0.1 | 1.7 | 262.6 | 47113.4 | <0.1 | 0 | 0 | 0 | 0 | 0.1 | 90.1 | 16666.7 | <0.1 | 0.1 | 5.5 | 133.3 | <0.1 |
| Alvinellidae | | | | | | | | | | | | | | | | | | | | | | | | |
| *Paralvinella dela* | 0 | 0 | 0 | 0 | 0 | 0 | 0 | 0 | <0.1 | 0.7 | 126.4 | <0.1 | 0 | 0 | 0 | 0 | 0 | 0 | 0 | 0 | 0 | 0 | 0 | 0 |
| *Paralvinella palmiformis* | 0 | 0 | 0 | 0 | 0.1 | 10.4 | 5198.8 | 0 | <0.1 | 10.3 | 1854.2 | 0.1 | 3.3 | 726.4 | 386915.9 | 6.8 | 0.2 | 110.2 | 20370.4 | 0.8 | 0.7 | 43.6 | 1066.7 | 1.4 |
| *Paralvinella pandorae* | <0.1 | 0.7 | 141.6 | <0.1 | <0.1 | 2.5 | 1223.2 | <0.1 | 0.1 | 17.4 | 3118.4 | <0.1 | 0.1 | 17.6 | 9345.8 | <0.1 | 0.1 | 70.1 | 12963 | <0.1 | 5.4 | 354.3 | 8666.7 | <0.1 |
| *Paralvinella sulfincola* | 0 | 0 | 0 | 0 | 0.1 | 10.4 | 5198.8 | 0 | <0.1 | 6.6 | 1179.9 | 0.1 | 0.6 | 122.8 | 65420.6 | 0.5 | <0.1 | 10 | 1851.9 | <0.1 | 0.2 | 10.9 | 266.7 | 0 |
| Ampharetidae | | | | | | | | | | | | | | | | | | | | | | | | |
| *Amphisamytha carldarei* | 26.8 | 1932.1 | 389334.6 | 0.5 | 24.5 | 3367.5 | 1681651.4 | 0.5 | 34.8 | 5378.8 | 964938.9 | 1.8 | 0.4 | 80.7 | 42990.7 | <0.1 | 5.1 | 3004.6 | 555555.6 | 0.2 | 7.4 | 479.6 | 11733.3 | 0.1 |
| Ctenodrilidae | | | | | | | | | | | | | | | | | | | | | | | | |
| *Raricirrus* sp. | <0.1 | 0.2 | 47.2 | <0.1 | 0 | 0 | 0 | 0 | <0.1 | 1.9 | 337.1 | <0.1 | 0 | 0 | 0 | 0 | 0 | 0 | 0 | 0 | 0 | 0 | 0 | 0 |
| Spionidae | | | | | | | | | | | | | | | | | | | | | | | | |
| *Prionospio* sp. | <0.1 | 2.1 | 424.7 | <0.1 | 0 | 0 | 0 | 0 | <0.1 | 0.9 | 168.6 | <0.1 | 0 | 0 | 0 | 0 | 0 | 0 | 0 | 0 | 0 | 0 | 0 | 0 |
| **Mollusca** | | | | | | | | | | | | | | | | | | | | | | | | |
| **Aplacophora** | | | | | | | | | | | | | | | | | | | | | | | | |
| Simrothiellidae | | | | | | | | | | | | | | | | | | | | | | | | |
| *Helicoradomenia juani* | 3.7 | 263.7 | 53138.3 | <0.1 | 3.4 | 461.1 | 230275.2 | <0.1 | 5.6 | 861.9 | 154614.4 | 0.14 | 0.1 | 17.6 | 9345.8 | <0.1 | 1.3 | 791.2 | 146296.3 | <0.1 | 2 | 130.8 | 3200 | <0.1 |
| **Gastropoda** | | | | | | | | | | | | | | | | | | | | | | | | |
| Buccinidae | | | | | | | | | | | | | | | | | | | | | | | | |
| *Buccinum thermophilum* | 0.1 | 9.8 | 1982.1 | 3.2 | 0.1 | 18.4 | 9174.3 | 1.6 | <0.1 | 2.4 | 421.4 | 0.4 | 0 | 0 | 0 | 0 | 0.1 | 50.1 | 9259.3 | 5.8 | 0 | 0 | 0 | 0 |
| Provannidae | | | | | | | | | | | | | | | | | | | | | | | | |
| *Provanna variabilis* | 1.9 | 137 | 27607.4 | 2.2 | 1.6 | 224.8 | 112232.4 | 1.3 | 4.5 | 694.8 | 124652.3 | 6.8 | 0.3 | 61.4 | 32710.3 | 0.3 | 8.6 | 5117.8 | 946296.3 | 10 | 1.8 | 114.5 | 2800 | 0.4 |
| Peltospiridae | | | | | | | | | | | | | | | | | | | | | | | | |
| *Depressigyra globulus* | 10.8 | 779.9 | 157149.6 | 1.1 | 14.5 | 1986 | 991743.1 | 1.1 | 20.1 | 3105.4 | 557100.7 | 2.9 | 51.4 | 11294.7 | 6015887.9 | 13 | 44 | 26179.8 | 4840740.7 | 21 | 23.2 | 1515.1 | 37066.7 | 1.5 |
| Clypeosectidae | | | | | | | | | | | | | | | | | | | | | | | | |
| *Clypeosectus curvus* | 0.5 | 34.7 | 6984.4 | 0.1 | 0.2 | 20.2 | 10091.7 | <0.1 | <0.1 | 6.8 | 1222.1 | <0.1 | 0 | 0 | 0 | 0 | 0.1 | 50.1 | 9259.3 | <0.1 | 0 | 0 | 0 | 0 |
| Lepetodrilidae | | | | | | | | | | | | | | | | | | | | | | | | |
| *Lepetodrilus fucensis* | 42.7 | 3083.1 | 621283.6 | 12.5 | 39.5 | 5429.4 | 2711315 | 10.1 | 22.8 | 3519.5 | 631394.9 | 11.5 | 43 | 9457.6 | 5037383.2 | 19.4 | 30.1 | 179907.2 | 3311111.1 | 18.1 | 55 | 3586.1 | 87733.3 | 18.2 |
| **Arthropoda** | | | | | | | | | | | | | | | | | | | | | | | | |
| **Arachnida** | | | | | | | | | | | | | | | | | | | | | | | | |
| Halacaridae | | | | | | | | | | | | | | | | | | | | | | | | |
| *Copidognathus papillatus* | 4.9 | 349.6 | 70457.8 | <0.1 | 1.5 | 211.9 | 105810.4 | <0.1 | 4.9 | 754.7 | 135398.2 | <0.1 | <0.1 | 8.8 | 4672.9 | <0.1 | <0.1 | 60.1 | 11111.1 | <0.1 | 0 | 0 | 0 | 0 |
| **Amphipoda** | | | | | | | | | | | | | | | | | | | | | | | | |
| Alicellidae | | | | | | | | | | | | | | | | | | | | | | | | |
| *Paralicella* cf. *vaporalis* | <0.1 | 1.6 | 330.3 | <0.1 | <0.1 | 6.1 | 3058.1 | <0.1 | <0.1 | 3.1 | 547.8 | <0.1 | 0 | 0 | 0 | 0 | 0 | 0 | 0 | 0 | 0 | 0 | 0 | 0 |
| Calliopiidae | | | | | | | | | | | | | | | | | | | | | | | | |
| *Oradarea* cf. *walkeri* | <0.1 | 4.7 | 943.8 | <0.1 | 0 | 0 | 0 | 0 | 0 | 0 | 0 | 0 | 0 | 0 | 0 | 0 | 0 | 0 | 0 | 0 | 0 | 0 | 0 | 0 |
| *Letpamphopus* sp. | 0 | 0 | 0 | 0 | 0 | 0 | 0 | 0 | 0 | 0 | 0 | 0 | <0.1 | 3.5 | 1869.2 | <0.1 | 0 | 0 | 0 | 0 | 0 | 0 | 0 | 0 |
| **Crustacea** | | | | | | | | | | | | | | | | | | | | | | | | |
| Philomedidae | | | | | | | | | | | | | | | | | | | | | | | | |
| *Euphilomedes climax* | 0.7 | 52 | 10476.6 | <0.1 | 10.9 | 1502.2 | 750152.9 | 0.1 | 0.2 | 23.3 | 4171.9 | <0.1 | 0.3 | 70.2 | 37383.2 | <0.1 | 9.1 | 5378.2 | 994444.4 | <0.1 | 3.3 | 218 | 5333.3 | <0.1 |
| Cytheruridae | | | | | | | | | | | | | | | | | | | | | | | | |
| *Xylocythere* sp. nov. | 2.9 | 206.8 | 41670.6 | 0.1 | 1.4 | 188 | 93883.8 | <0.1 | 2.7 | 413 | 74083.4 | <0.1 | <0.1 | 7 | 3738.3 | <0.1 | 0.3 | 170.3 | 31481.5 | <0.1 | 0.1 | 5.5 | 133.3 | <0.1 |
| **Pycnogonida** | | | | | | | | | | | | | | | | | | | | | | | | |
| Ammotheidae | | | | | | | | | | | | | | | | | | | | | | | | |
| *Sericosura verenae* | 0.6 | 42.6 | 8589.0 | <0.1 | 0.6 | 84.5 | 42201.8 | 0.5 | 0.7 | 113.5 | 20354 | 0.8 | 0.1 | 24.6 | 13084.1 | 0.2 | 0.6 | 360.6 | 66666.7 | 1.2 | 0.4 | 27.3 | 666.7 | 0.2 |
| *Sericosura venticola* | 0.2 | 15.2 | 3067.5 | <0.1 | 0.1 | 8 | 3975.5 | 0.5 | <0.1 | 1.9 | 337.1 | 0.5 | <0.1 | 1.8 | 934.6 | <0.1 | 0 | 0 | 0 | 0 | 0 | 0 | 0 | 0 |
| *Sericosura* cf. *dissita* | 0.1 | 5.6 | 1132.6 | <0.1 | <0.1 | 1.2 | 611.6 | 0.5 | <0.1 | 0.7 | 126.4 | 0.1 | <0.1 | | | <0.1 | 0 | 0 | 0 | 0 | 0 | 0 | 0 | 0 |
| **Nemertea** | | | | | | | | | | | | | | | | | | | | | | | | |
| Unidentified | <0.1 | 1.2 | 236.0 | <0.1 | | | | | <0.1 | 2.8 | 505.7 | <0.1 | 0 | 0 | 0 | 0 | 0 | 0 | 0 | 0 | 0 | 0 | 0 | 0 |
| **Echinodermata** | | | | | | | | | | | | | | | | | | | | | | | | |
| Ophiuroidea | <0.1 | 0.2 | 47.2 | <0.1 | | | | | <0.1 | 0.5 | 84.3 | <0.1 | 0 | 0 | 0 | 0 | 0 | 0 | 0 | 0 | 0 | 0 | 0 | 0 |

**Table 3.** Univariate measures of macrofaunal community structure associated with *Ridgeia piscesae* tubeworm bushes on the Grotto edifice: Sample Surface Area (SSA), Tube Height (TH), Tube Diameter (TD), volume (V), species richness (S), exponential of Shannon entropy (D), Simpson's diversity index (1-λ') and Pielou's evenness (J').

| Sample | SSA (m$^2$) | TH (cm) ± sd | TD (cm) ± sd | TSA (m$^2$) | V (m$^3$) | S | D | 1-λ' | J' |
|--------|--------|--------------|--------------|---------|--------|----|-------|-------|-------|
| S1 | 0.12 | 17.14 ± 6.83 | 4.72 ± 0.82 | 4.27 | 0.02 | 28 | 5.377 | 0.728 | 0.505 |
| S2 | 0.06 | 5.32 ± 2.47 | 2.24 ± 0.42 | 1.63 | $3.4 \times 10^{-3}$ | 24 | 5.398 | 0.749 | 0.531 |
| S3 | 0.12 | 19.91 ± 9.31 | 5.48 ± 1.13 | 4.26 | 0.02 | 31 | 6.053 | 0.778 | 0.524 |
| S4 | 0.02 | 7.15 ± 2.45 | 2.48 ± 0.57 | 0.57 | $1.1 \times 10^{-3}$ | 19 | 2.605 | 0.55 | 0.325 |
| S5 | 0.02 | 3.46 ± 0.85 | 1.91 ± 0.34 | 0.10 | $5.4 \times 10^{-4}$ | 19 | 4.348 | 0.697 | 0.499 |
| S6 | 0.01 | 6.15 ± 2.80 | 2.14 ± 0.43 | 0.18 | $7.5 \times 10^{-3}$ | 14 | 3.998 | 0.633 | 0.525 |